



# Sediment shell-content diminishes current-driven sand ripple development and migration

Chiu H. Cheng[1], Jaco C. de Smit[1,2], Greg S. Fivash[1], Suzanne J. M. H. Hulscher[3], Bas W. Borsje[3], Karline Soetaert[1]

5  [1]NIOZ Royal Netherlands Institute for Sea Research, Department of Estuarine and Delta Systems (EDS), 4400 AC Yerseke, The Netherlands.
[2]Faculty of Geosciences, Department of Physical Geography, Utrecht University, 3584 CB Utrecht, The Netherlands
[3]Water Engineering and Management, University of Twente, 7500 AE Enschede, The Netherlands.

10  *Correspondence to*: Chiu H. Cheng (chiu.cheng@nioz.nl)



**Abstract.** Shells and shell fragments are biogenic structures that are widespread throughout natural sandy shelf seas and whose presence can affect the bed roughness and erodibility of the seabed. An important and direct consequence is the effect on the formation and movement of small bedforms such as sand ripples. We experimentally measured ripple formation and migration of a mixture of natural sand with increasing volumes of shell material in a racetrack flume. Our experiments reveal the impacts of shells on ripple development in sandy sediment, providing information that was previously lacking. Shells expedite the onset of sediment transport while simultaneously reducing ripple dimensions and slowing down their migration rates. Moreover, increasing shell content enhances near-bed flow velocity due to the reduction of bed friction that is partly caused by a decrease in average ripple size and occurrence. This, in essence, limits the rate and magnitude of bedload transport. Given the large influence of shell content on sediment dynamics on the one hand, and the high shell concentrations found naturally in the sediments of shallow seas on the other hand, a significant control from shells on the morphodynamics of sandy marine habitats is expected.



## 1 Introduction

Ripples are the most common bedforms found in the marine environment, including in shallow, sandy environments (Bartholdy et al., 2015; Langlois and Valance, 2007). They form over a broad range of sandy grain mixtures under low energy flow or wave conditions that exceed the erosion threshold (Precht and Huettel, 2003; Soulsby, 1997). With increasing water depth, ripples become progressively driven by currents rather than waves. Current-generated ripples are very dynamic microscale bedforms, with typical sizes of around 1 m in wavelength and up to 0.01 m or more in height (Knaapen et al., 2005; van Rijn et al., 1993). They continuously develop and erode, typically on the order of minutes to days, and can migrate at rates exceeding 0.4 cm min$^{-1}$ (Baas et al., 2000; Baas and De Koning, 1995; Bartholdy et al., 2015; Lichtman et al., 2018; Miles et al., 2014). As the ripples move and change in dimension, the bed roughness is correspondingly altered, which can have cascading effects on the surrounding areas such as larger bedforms (e.g., tidal sand waves) on which they are often superimposed (Brakenhoff et al., 2020; Damveld et al., 2018, 2019; Idier et al., 2004). Additionally, ripples also generate distinct spatial variations in sediment composition and alter the distribution of particulate organic matter by their effect on hydrodynamics, some of which can further modify the sediment properties in ways that influence erosion (Ahmerkamp et al., 2015; Kösters and Winter, 2014; Lichtman et al., 2018; Malarkey et al., 2015; Mietta et al., 2009).

Shells, a biogenic material created by marine bivalves, are widely distributed in certain regions of the marine environment (Russell-Hunter, 1983). These calcareous structures remain present long after the death of the organisms (Gutiérrez et al., 2003; Kidwell, 1985), and they are mostly found in the form of separated single shell valves and shell fragments. In environments where shells are prevalent, they may constitute 20–70 % of the total sediment composition (by volumetric percentage), although even higher percentages have been observed in very extreme cases (Dey, 2003; Soulsby, 1997). Since they have a lower bulk density, their presence reduces the bulk density of the sediment by diluting the quartz fraction (Soulsby, 1997). As shell material is rather plate-like, irregular and angular in shape, they also change the general profile compared to the smaller, surrounding sediment particles (Al-Dabbas and McManus, 1987). Intact shells and larger fragments may inhibit sediment transport through bed armoring. Armoring occurs when the mean shear stress is below the critical erosion threshold for the coarsest fractions, but above that for the finer particles, resulting in their entrainment. This winnowing causes the surface to become coarser and coarser, essentially building up an armor layer (Vericat et al., 2006). In riverine environments, coarse material such as gravel has been shown to facilitate bed armoring, causing the upper layers of the sediment to become significantly coarser than the median grain size ($D_{50}$) of the sediment beneath, ultimately reducing or inhibiting sediment transport (Curran, 2010; Wilcock and Detemple, 2005). Shells may also be able to provide a similar armoring effect against sand erosion given that they are more difficult to erode (Miedema and Ramsdell, 2011; Ramsdell and Miedema, 2010).

Thus far, very few studies have investigated the direct influence of shell material on the bedload transport dynamics through the alteration of bed roughness (Gutiérrez et al., 2003; Nowell and Jumars, 1984). Some studies have explored the ways in which shells could be used as tracers for sediment motion, given their widespread occurrence (Al-Dabbas and



McManus, 1987). The drag and incipient motion of the valves of a few bivalve species have also been investigated in the laboratory (Dey, 2003). Similar studies have focused on the erosion and settling velocities of shells, based on shapes, shell positioning and associated drag, being transported through a pipeline (Miedema and Ramsdell, 2011; Ramsdell and Miedema, 2010). Although these studies have considered how the irregularity in shape and orientation of shell valves potentially interact with flow, the focus has been more within a hydrodynamic context rather than a sedimentary one. To our knowledge, there have not been studies addressing the direct effects of a natural, representative mixture of shells and sandy sediment on the development and movement of ripples.

On the one hand, shells behave differently than rock and other inorganic fragments of similar size in that they are hydraulically somewhat more similar to siliciclastic particles, even when the sizes differ greatly (Al-Dabbas and McManus, 1987). On the other hand, due to the shape and size of most shells, combined with their lower density, they are known to have a much lower settling velocity and much larger erosion velocity threshold than sand particles (Ramsdell and Miedema, 2010). The mere presence of dead shells has also been shown to facilitate silt and other fine-particle entrainment in the sediment (Huettel and Rusch, 2000; Pilditch et al., 1997; Witbaard et al., 2016). But despite their prevalent nature and potential to affect sediment dynamics in several ways, there is at present a knowledge gap in terms of the direct influence of shells on the geomorphology of sandy sediments.

The objective of this experiment was to determine the effect of biogenic shells on the development of ripples in fine sand, in relation to unidirectional flow and turbulence along the bed. The combination of flow velocity and turbulence intensity largely dictates the sediment dynamics, thereby affecting bedform development and conditions within the sediment (Blanchard et al., 1997; Herman et al., 2001; Paterson et al., 2001). Bottom roughness and small-scale topography are important contributing factors to bedform pattern development (Van Oyen et al., 2010), and past studies have found a significant effect of epibenthic structures at different densities (e.g., mimics of tube worm reef patches) on flow and sediment erosion (Friedrichs et al., 2000, 2009). However, the influence of shell material on the ripple dynamics, in relation to flow and turbulence, is still not well understood. Thus, we aimed to quantify the turbulence generated by the flow along sand ripples, simulating scenarios both with and without the presence of shells. We used shells from common bivalve species found in the sandy Dutch North Sea including *Spisula* spp., *Tellimya* spp. and *Cerastoderma edule*, at increasing densities. Using dead shells, we determined the influence, via autogenic engineering, of shell material on sediment transport by testing the effects of increasing shell content on ripple formation, shape and migration rates.

Our paper is organized as follows. The experimental setup, instrumentation and analyses utilized are described in section 2. The results, including the incipient sediment motion, ripple migration and other ripple calculations are presented in section 3. The significance of our findings are discussed in section 4. In section 5, the final conclusions are presented.





## 2 Materials and Methods

### 2.1 The experimental setup

Our experiments were performed in a racetrack flume facility located at the NIOZ Royal Netherlands Institute for Sea Research, Yerseke, The Netherlands. This large, unidirectional flow channel measures approximately 17.5 m in length and 3.25 m in width and can generate depth-averaged currents up to 60 cm s$^{-1}$ (Figure 1a). A test section containing a sediment basin measuring 200 x 60 x 25 cm (L x W x H) is located at the far end of one of the long, straight sections of the flume to minimize the effect of bend flows (Figure 1b). The drive belt equipped on the backside allows the flow to be controlled with high precision.

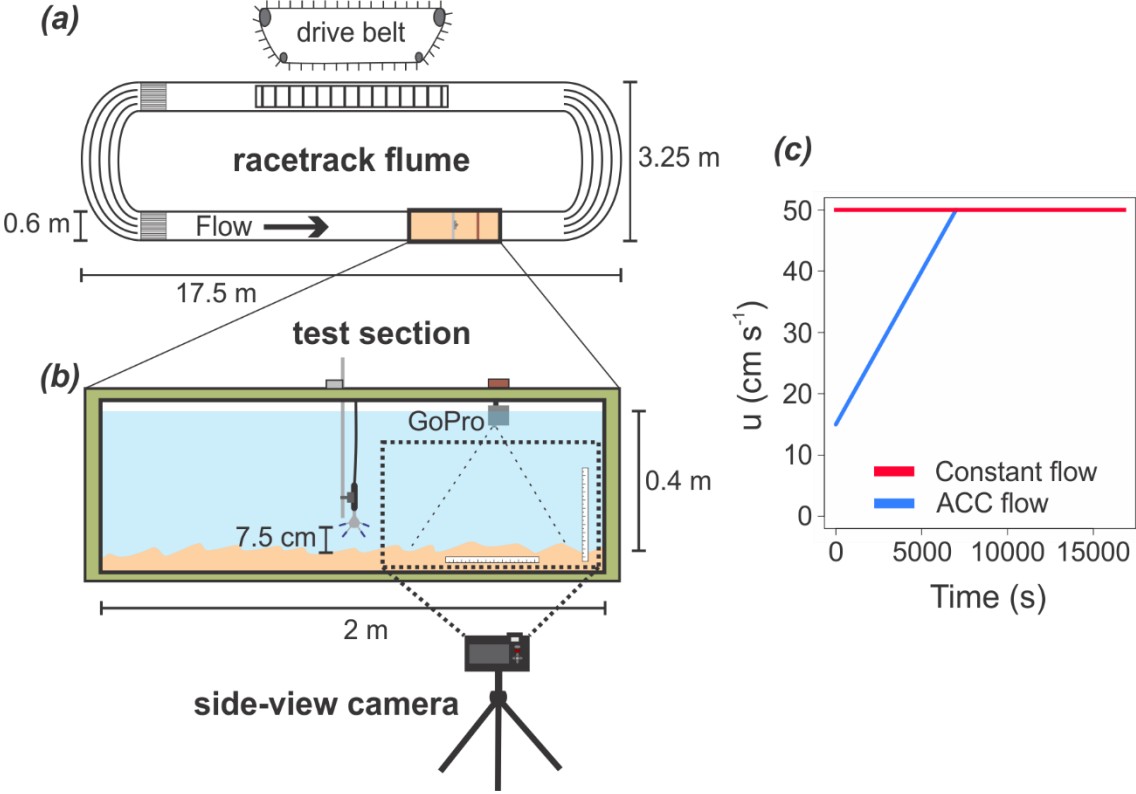

**Figure 1:** *(a)* **The top view of the NIOZ racetrack flume.** *(b)* **A side view of the 2 m long test section showing the ADV vectrino, GoPro and side-view cameras used in each experimental run. Both cameras were positioned within the 2$^{nd}$ half of the viewing window of the test section.** *(c)* **The flume flow settings implemented in the two separate experiments.** *Note*: **Although ripples are shown for illustrative purposes, the indicated water depth and ADV height are based on the initial (flat bed) condition.**

The basin of the 2 m test section was filled using North Sea sandy sediment (see Table S1 for the properties). To maintain a sediment supply throughout the duration of each individual experiment, a thin layer of sand (~3 cm) was also placed





over the 3 m preceding the measurement section of the flume track. The bed was fully mixed and flattened before each experimental run. The total water depth was 40 cm and only freshwater was used.

For the shell treatments, we used a mixture that consisted of, on average, approximately 29 % intact shells valves and 71 % fragments (in absolute number of pieces). All non-shell materials (e.g., rocks and wood detritus) were removed prior to
the addition. We took a random sampling of the shell stockpile to determine the average dimensional properties of the shell valves and fragments (Table S2; Fig. S1).

Two separate experiments were conducted. A constant flow experiment was used to measure equilibrium ripple dimensions and migration rates. An acceleration (ACC) flow experiment was run to measure the incipient sediment motion. The flow settings used in the two experiments are shown in Figure 1c. Both experiments consisted of several experimental
runs, which were varied by changing the volumetric percentage of shell content. The control (0 % shells) contained only sandy sediment, while each subsequent treatment was modified by the addition of shell material. The volumetric percentage of shell increased by 2.5 or 5 % intervals, up to 30 %, while the last two treatments contained 40 % and 50 % shells, respectively. The flume was filled with water overnight, and all experimental runs were always performed the very next day to maintain consistency (e.g., minimize variability due to compaction, etc.). The constant flow experiment consisted of six treatments (0,
5, 10, 15, 20, and 50 % shell), while the ACC flow experiment included 11 treatments (0, 2.5, 7.5, 10, 12.5, 15, 20, 25, 30, 40 and 50).

## 2.2 Constant flow experiment

In the constant flow experimental runs, a 50 cm s$^{-1}$ depth-averaged flow velocity was maintained for more than 4 hours, so as to achieve equilibrium conditions (Figure 1c). Preliminary runs showed that morphological equilibrium was achieved well
within one hour at this flow rate. A Canon EOS 1000D camera, equipped with an EX Sigma lens (DG Macro, 50 mm, 1:2.8) was positioned at the side of the flume, targeting the 2$^{nd}$ half of the test section to record time-lapse photos from the side at 10-second intervals. The photos recorded a section 76.5 cm in width and 51 cm in height. Two rulers were attached at the edges of the frame as dimensional guides for the image analyses (Figure 1b).

Concurrently, a Nortek Vectrino ADV profiler was used to record the 3-dimensional flow rates, through coherent
Doppler processing, at a frequency of 30 Hz. Data was filtered for minimum correlation values of 90 %, minimum signal-to-noise ratio of 20 dB and minimum amplitude of -35 dB. The probe was placed approximately 7.5 cm above the bed, which was initially flat in each experimental run. With a blanking distance of 4 cm, it measured the bottom section of the water column from 0 to 3.5 cm above the initial flat bed, over a total of 35 cells (1 mm intervals). The ADV was held in place through the duration of the experimental run. Therefore, near-bed flow profiles were corrected for changing bed elevation during ripple
migration.



## 2.3 Sediment image processing

Identification of ripples within the sediment bed was performed through image analysis of the photo time-series obtained by the camera. The vertical position of the sediment-water interface was identified using Canny edge detection of the green band with the *wvtool* package (Sugiyama and Kobayashi, 2016), which showed highest contrast. Gamma transformation of the green band further enhanced this contrast to improve the quality of the detection. The fine-grain noise in the sediment surface was filtered out using a low-pass 2nd-order Butterworth filter to produce a smooth surface from which peaks and troughs can be easily identified, using the *signal* package (Ligges et al., 2015). Ripples were then classified from the identified sediment surface using peak analysis, which isolated peaks and troughs in the sediment surface with the *pracma* package (Borchers, 2019). This ultimately allowed us to characterize the dimensions of individual ripples and track their movement and development in time. Using 1600 unique frames from each of the six constant flow experimental runs, we quantified the following ripple parameters: (1) the ripple height, (2) length, (3) asymmetry and (4) migration rate.

Each ripple was defined as encompassing the region between two neighboring troughs, separated by a peak. The ripple height and length were defined as the maximum vertical and lateral extent of the ripple. The ripple asymmetry was defined as the difference in length between the two halves of the ripple, separated by the center of its peak, divided by its total length (trough-trough); values change from 0 (highest symmetry) to 1 (highest asymmetry). The migration rate was calculated as the total distance traveled by the peak of a unique ripple over 24 frames (constituting an interval of four minutes). This frame interval allowed ripples to travel measurable distances while limiting the likelihood of them moving out of frame before measurements could be taken. All image analyses were conducted in R version 3.4.4 (R Core Development Team 2020).

## 2.4 Near-bed flow calculations

The near-bed turbulent kinetic energy (TKE) was derived from the near-bed flow velocity fluctuations (Pope et al., 2006). This value indicates the mean kinetic energy associated with eddies from the turbulent flow. The near-bed TKE was calculated from near-bed flow velocity fluctuations in the x, y and z directions as:

$$TKE = 1/2 \left( \overline{u'_{b,x}}^2 + \overline{u'_{b,y}}^2 + \overline{u'_{b,z}}^2 \right) \tag{1}$$

where $\overline{u'_{b,x}}$, $\overline{u'_{b,y}}$ and $\overline{u'_{b,z}}$ represent the root-mean-squares of the near-bed flow velocity fluctuations in the x, y and z directions, respectively. These values were extracted from the flow velocity signal through means of applying a 0.1 Hz high-pass 5th order Butterworth filter. This ensures removal of the background velocity during the measurement period. Another 10 Hz low-pass 5th order Butterworth filter was used to remove the higher frequencies where the signal was dominated by noise. The corresponding bottom shear stress (BSS) was calculated as (Soulsby, 1983):

$$BSS = 0.19\rho TKE \tag{2}$$



Where $\rho$ is the water density (1000 kg m$^{-3}$ for freshwater). Subsequently, the corresponding effective bed roughness, which is
affected by both shells and bed forms, can be calculated from the depth-averaged velocity and the BSS. For a unidirectional
flow, the BSS can be calculated from the depth-averaged velocity as (van Rijn, 1993):

$$BSS = \rho g u^2 / C^2 \tag{3}$$

Where $u$ is the depth-averaged velocity (m s$^{-1}$), $g$ is the gravitational acceleration (9.81 m s$^{-2}$) and $C$ is the Chézy roughness
coefficient (m$^{0.5}$ s$^{-1}$). The Chézy roughness coefficient is a function of the water depth and bed roughness (van Rijn, 1993):

$$C = 18 \log(12h/ks) \tag{4}$$

Where $h$ is the water depth (0.4 m in this flume experiment), and $ks$ is the effective bed roughness (m).

## 2.5 ACC flow experiment

This experiment was conducted to measure the onset of incipient sediment transport, as well as the corresponding boundary
layer conditions. Incipient sediment transport was measured for a flat bed configuration in order to quantify the direct effect
of shells on sediment stability. Sandy sediment with a $D_{50}$ of 350 μm is not expected to exhibit sediment motion below about
30 cm s$^{-1}$ (van Rijn, 1993), and an initial test run with our setup showed that there was indeed no sediment movement occurring
below 20 cm s$^{-1}$. Thus, the starting velocity of each run was set at 15 cm s$^{-1}$. The flow speed was linearly increased at a rate of
0.3 cm s$^{-1}$ per minute from 15 to 50 cm s$^{-1}$ (over a time frame of 116.6 minutes).
       The ADV profiler was again anchored in the middle of the test section. One GoPro Hero3 camera was positioned just
below the water surface, looking downward, 1.5 m along the test section to produce top-view video recordings of the sediment
surface at 2 frames per second. The onset of incipient motion, which is the movement of a sufficient amount of particles to
result in a significant change in bed configuration, was derived visually from the GoPro footage. Visual observation is an
accurate method to determine erosion thresholds. As bed load transport is proportional to flow velocity to the power of 3, a
small change in velocity will lead to a significant and well-observable change in sediment transport. The depth-averaged
velocity was determined from the flume setting at the identified time when incipient motion was observed (Figure 1c). The
critical mean near-bed flow velocity, TKE and BSS were derived from the ADV measurements over the 5 minutes preceding
and 5 minutes after the onset of incipient motion following Equations (1) and (2). The effective bed roughness for flat beds
with varying shell content was calculated following Equations (3) and (4), using a 10-minute window of ADV measurements
at an average flow rate of 20 cm s$^{-1}$, before any ripples had formed.





## 3 Results

We tested a large range in the shell content in our constant flow experiment, and the results clearly demonstrate that the development of ripples is strongly controlled by the shell fraction of sandy sediments. Consequently, the ripple height, length, asymmetry and migration rate were all affected by shell content, and a drastic change in ripple development can clearly be seen in the concluding frames of each experimental run, particularly around 20 % in the constant flow experiment (Figure 2a).

While these ripple parameters were not measured in the ACC flow experiment, a similar observation could still be seen at around 15 % shell content (Figure 2b). In addition, as the shell percentage increased in the experimental runs, they began to form larger, aggregated clusters. These bands of shells were rather immobile, and the already-smaller ripples were observed from the GoPro videos to either migrate around the larger and highest clusters or disappear altogether (Figure 3).



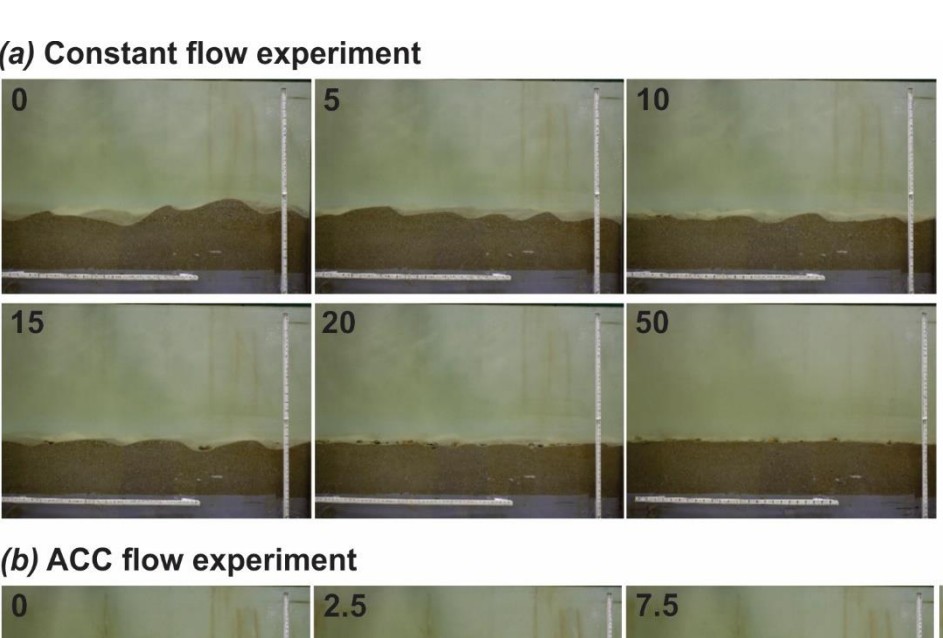

**Figure 2: The final frame from each** *(a)* **Constant flow and** *(b)* **ACC flow experimental run. Numbers represent the shell %. The white vertical and horizontal rulers are both 50 cm in length.**



Earth **Surface**
**Dynamics**
Discussions



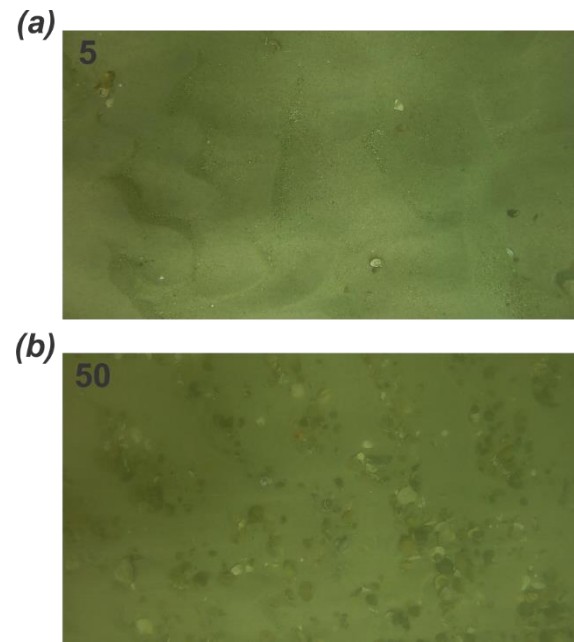

**Figure 3: Stills taken from the GoPro videos (constant flow experiment) to show the contrast between a** *(a)* **low-density and** *(b)* **high-density treatment. Shells increasingly form clusters at higher concentrations. The numbers represent the shell %.**

## 3.1 Changes to ripple characteristics (constant flow experiment)

An increase in the shell percentage reduced the spatial dimensions of the ripples.. Overall, the lengths and heights of the ripples decreased with increasing shell content (Figure 4a and b). Ripple asymmetry exhibited a slight decrease with increasing shell content. The migration rate showed a consistent decrease with increasing shell content (Figure 4c and d).



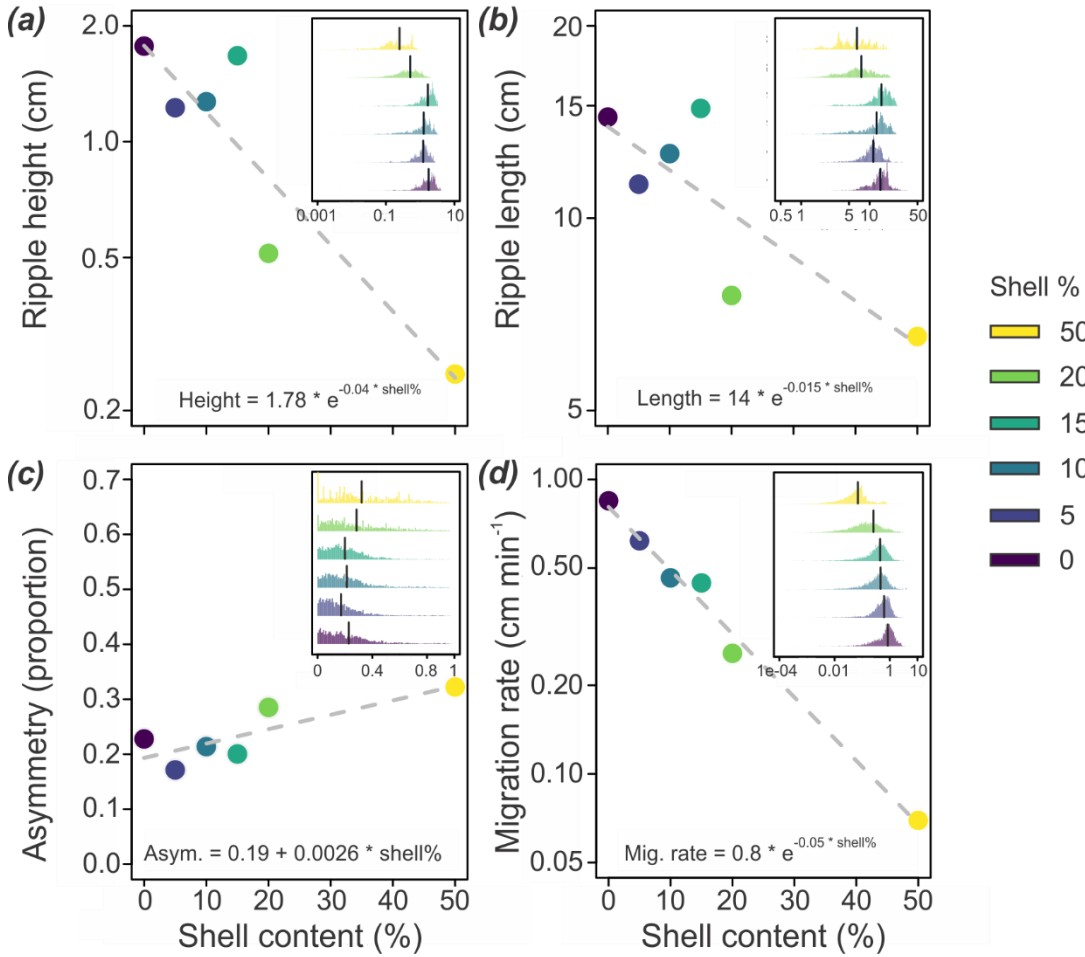

**Figure 4:** *(a)* **Ripple height,** *(b)* **Ripple length,** *(c)* **Asymmetry and** *(d)* **Migration rate, plotted against the shell content from the constant flow experimental runs. The y-axis of the ripple height, length and migration are plotted under a log scale.** *Inset panels:* **The corresponding histograms for each ripple parameter, with the x-axis values representing the y-axis values of the respective regression plots. Vertical lines represent mean values.**

The average ripple height decreased at a rate of -0.03 cm shell %$^{-1}$ ($F_{1,4}$ = 19.16, $R^2$ = 0.784, n = 6, p < 0.0001), from 1.77 ± 0.013 (mean ± se) cm (0 % shell) to 0.25 ± 0.007 cm (50 % shell) (Figure 4a). The average length of the ripples also decreased at a rate of -0.16 cm shell %$^{-1}$, from 14.4 ± 0.07 cm (0 %) to 6.53 ± 0.04 cm (50 %), but this change was less consistent between each of the experimental runs ($F_{1,4}$ = 7.8, $R^2$ = 0.58, n = 6, p = 0.049) (Figure 4b). The ripple symmetry varied only marginally as a consequence of shell content ($F_{1,4}$ = 6.5, $R^2$ = 0.52, n = 6, p = 0.06), increasing in asymmetry with increasing shell material from 0.23 ± 0.0032 (0 % shell) to 0.32 ± 0.0019 (50 % shell), at an average rate of 0.002 shell %$^{-1}$ (Figure 4c). The ripple migration rate was strongly affected by shell content, slowing at an average rate of -0.016 cm min$^{-1}$





shell %$^{-1}$, which reduced migration by an order of magnitude between the 0 % and 50 % shell treatments, from 0.85 ± 0.009 cm min$^{-1}$ to 0.07 ± 0.005 cm min$^{-1}$ ($F_{1,4}$ = 295.2, $R^2$ = 0.983, n = 6, p < 0.0001; Figure 4d).

### 3.2 Changes to near-bed hydrodynamics and critical BSS

In the constant flow experiment, the presence of shells (at all percentages) enhanced the near-bed flow in the horizontal direction (Figure 5a), as ripple sizes become diminished (Figure 4). Near-bed vertical flow was on average directed downwards, and reduced towards increasing shell content (Figure 5b). Interestingly, while the increasing near-bed flow
velocity with increasing shell percentages indicates a reduction in overall bed friction (Figure 5a), the highest TKE is observed at 50 % shell content (Figure 5c).



Earth **Surface**
**Dynamics**
Discussions

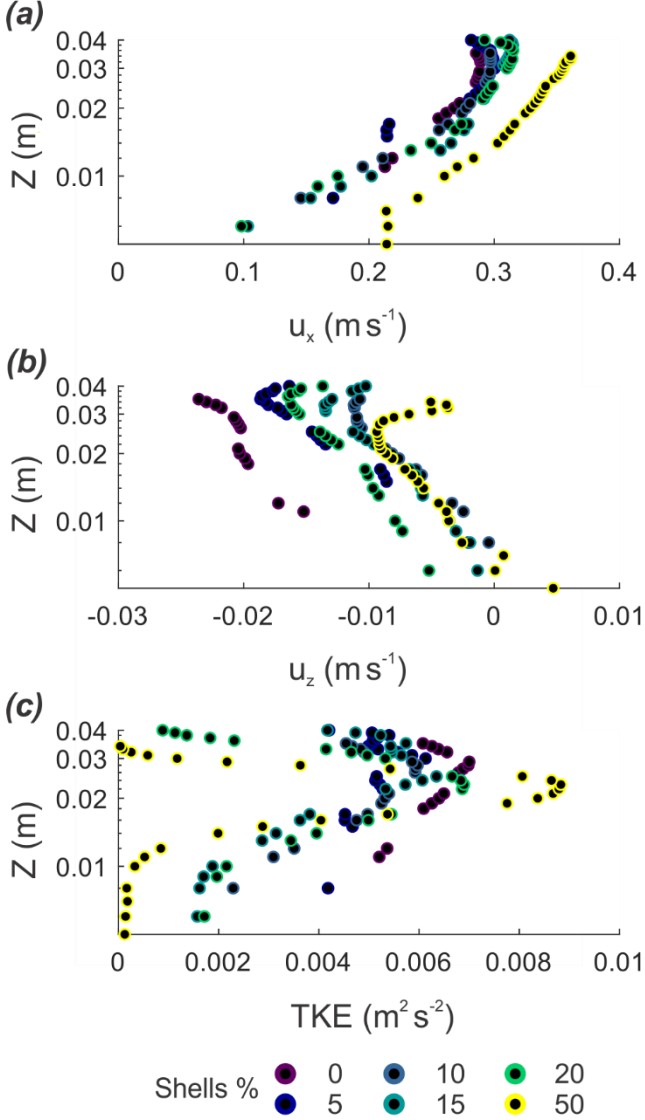

**Figure 5: Time-averaged near-bed velocity profiles showing the (*a*) x and (*b*) z direction of the constant flow experimental runs, as well as the (*c*) the TKE profiles. *Note*: The profiles are time-averaged over the entire duration of each experimental run.**

The critical near-bed velocity profiles from the ACC flow experimental runs showed a large reduction in critical near-bed velocity between 0 and 15 % shell content, followed by a minor reduction towards the 50 % shell content (Figure 6a). No differences were observed between the vertical velocity profiles (Figure 6b), which averaged 0 as the ripples were absent. Shells had a strong influence on the critical TKE and BSS (Figure 6c and 7a). The addition of shell material initially increased

the critical BSS from approximately 0.2 N m$^{-2}$ at 0 % shell content to approximately 0.75 N m$^{-2}$ at 2.5 % shell content (Figure 7a). Subsequently, the critical BSS dropped towards 0.25 N m$^{-2}$ at 15 % shell content ($R^2$ = 0.91, Figure 7a). At shell





concentrations above 20 %, the critical BSS slowly increased again to approximately 0.5 N m$^{-2}$ at 50 % shell content ($R^2$ = 0.50, Figure 7a). In contrast to the critical BSS, the critical depth-averaged velocity for incipient motion consistently reduced towards 15 % shell content ($R^2$ = 0.99, Figure 7b), after which it stayed constant ($R^2$ = 0.29, Figure 7b).

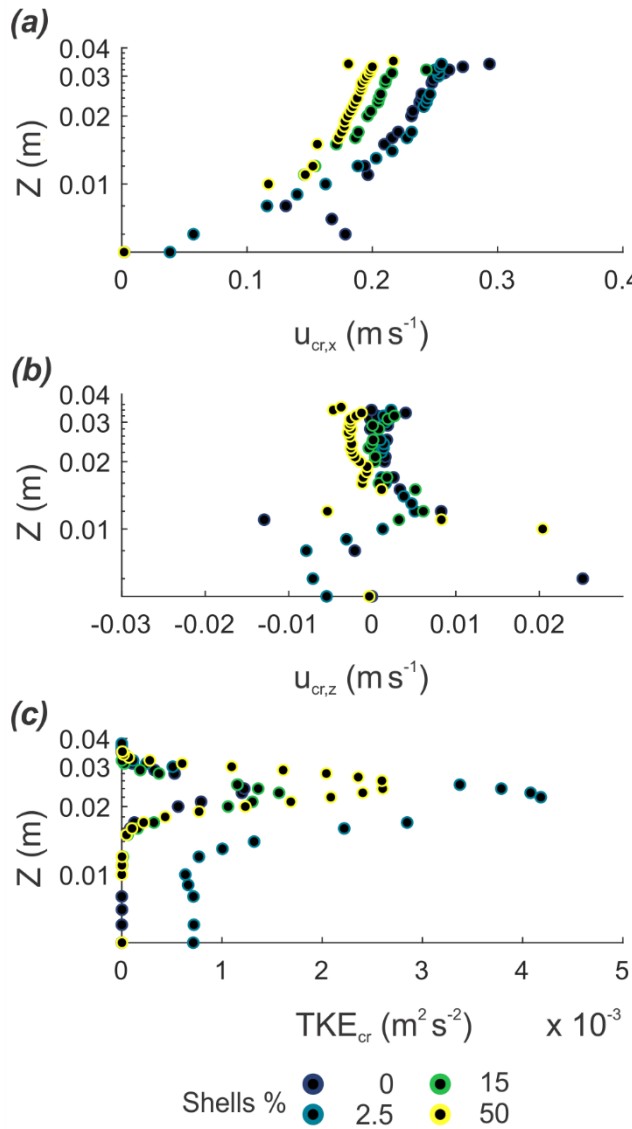

**Figure 6:** *(a)* **Near-bed horizontal flow,** *(b)* **vertical flow and** *(c)* **TKE profiles at the onset of sediment transport for flat beds (ACC flow experiment).** *Note*: **The profiles are time-averaged over a 10-minute period, which encompasses the 5 minutes prior to and following the incipient motion, for the four selected experimental runs.**

Earth **Surface**
**Dynamics**
Discussions

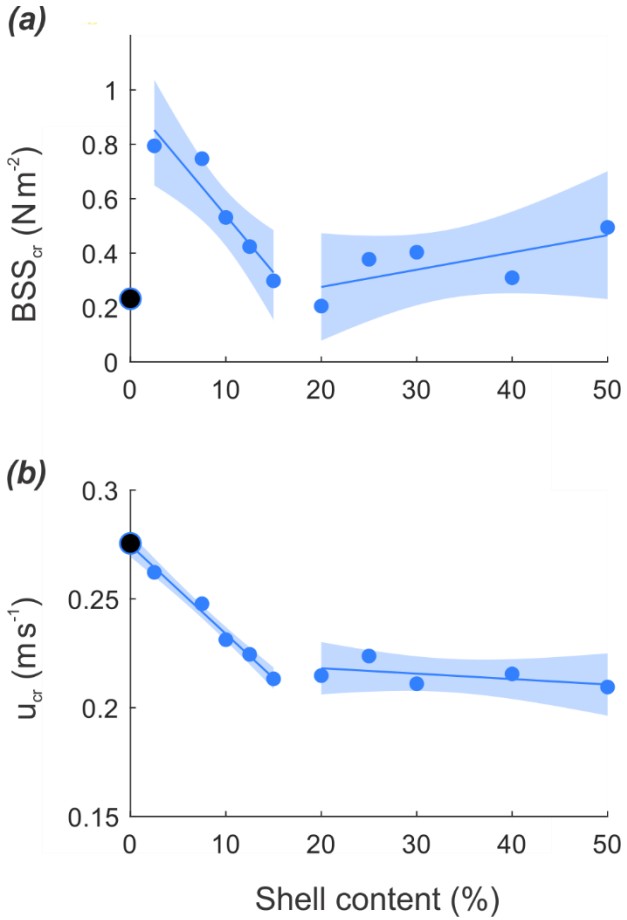

**Figure 7:** *(a):* **The critical BSS for incipient motion.** *(b)* **The corresponding depth-averaged velocity. The shaded regions represent the 95th percentile confidence intervals.**

The influence of shells on the effective bed roughness showed contrasting behavior between flat (e.g., ACC flow experiment) and equilibrium (e.g., constant flow experiment) beds (Figure 8). Under the absence of bed forms, the effective

bed roughness showed a similar trend as the critical BSS; a large increase from $1.2 \times 10^{-4}$ m to 0.042 m between 0 and 7.5 % shell content, followed by a decrease to 0.007 m at 15 % shell content, after which it stabilized at 0.005 ±0.004 m towards 50 % shell content. When ripples were present (equilibrium bed), the effective bed roughness decreased from 0.02 to 0.013 m from 0 to 10 % shell content. Beyond 10 % shell content, the effective bed roughness increased to 0.036 m.





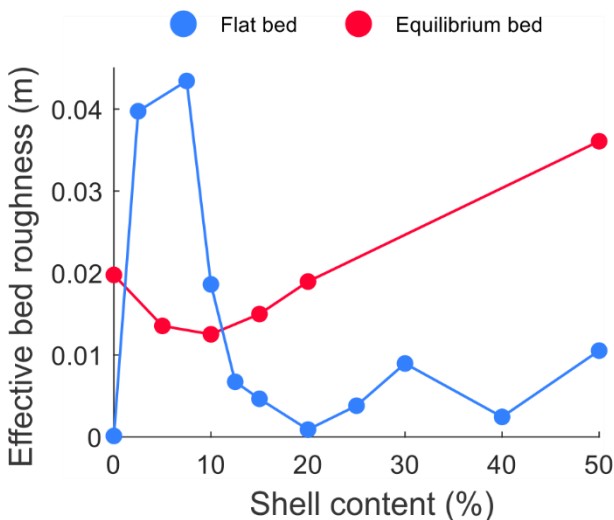

**Figure 8: Effective bed roughness against shell content for the flat beds (ACC flow experimental runs) and equilibrium beds (constant flow experimental runs).**

## 4 Discussion

The large range in the shell content that we tested in our experiments demonstrated that biogenic shells have a profound effect on the mechanical sediment properties in ways that could have ramifications for the bed evolution. By performing two types of measurements, we investigated both the (theoretical) equilibrium situation at constant high flow conditions (0.5 m s$^{-1}$), as well as the sequence of events that occur as the velocity increases (the ACC flow experiment). The latter pointed to the physical conditions under which sediment dynamics begin to change (e.g., incipient motion).

Under the constant flow conditions, all of the ripple parameters, except ripple symmetry, were highly affected by the presence of shells. The ripple height, length and migration rate all decreased exponentially as a function of shell content, such that the ripples almost entirely disappeared at 50 % shell content (Figure 2). The ripples also became slightly more asymmetric with increasing shell content (Figure 4c).

In contrast, the influence of shells on the initiation of bedload transport was much more complex. There appeared to be a threshold at ~ 15–20 % shells, marking a transition in the way the ripple movement was initiated. Above 20 % shells, the depth-averaged velocity that was required to initiate movement fluctuated around 0.22 m s$^{-1}$ (Figure 7b), while the critical BSS slightly increased with higher shell content (Figure 7a). Below 20 % shell content on the other hand, the depth-averaged velocity at which the sand started to move linearly decreased with shell content, from 0.27 m s$^{-1}$ for bare sand to 0.22 m s$^{-1}$ at 15 % shells. Intriguingly, the most immediate and drastic changes in the critical BSS occurred when the smallest quantity of shell was mixed into the sediment (2.5 %), which was enough to more than triple the critical BSS (Figure 7a). Between 2.5 and 15 % shells, the value of the *BSS* at which the sediment particles start to move decreased with increasing shell content.



### 4.1 Significance of shell-ripple interactions

In gravel bed rivers, it is known that the incorporation of structures into the sediment surface creates microclusters that increase both the bed roughness as well as bed stability (Curran, 2010). The anchoring of shells in sandy sediment greatly enhances their erosion threshold compared to individual shells situated on a flat surface, irrespective of the orientation. Whereas

individual or loose shells on top of a flat sandy surface can erode at velocities well below 0.4–0.5 m s$^{-1}$ (Dey, 2003), shells that are fixed in the sediment, especially in clusters, are much less susceptible to erosion. Despite our flow velocities reaching these thresholds in our experiments, the shells were mostly immobile, even as ripples migrated over them. In rare cases, the smaller valves and fragments sometimes shifted a few centimeters due to ripple movement. But in the higher shell treatments, the large shell clusters were practically fixed structures (Figure 3).

Therefore, a sandy sediment bed with sufficient quantity of shells under a unidirectional flow will produce an armoring effect somewhat similar to riverine environments, where gravel beds produce clustered structures that mediate the bed-flow interactions through a combination of bed stabilization, altered roughness and regulation of the amount of sediment available for transport (Curran, 2010; Tuijnder et al., 2009; Wilcock and Detemple, 2005). In addition, our experiments show that shell content has another indirect bed-mediating effect. Due to the dampening of the size of the ripples, and consequently

a reduction of the bottom roughness, there was a progressive enhancement of the mean near-bed flow (Figure 6) as a function of increased shell content, and a slowing down of the ripple migration rate (Figure 4d).

### 4.2 Shell vs. incipient sediment motion

The comparison of the critical depth-averaged velocity and BSS in the ACC flow experiment suggests that the influence of

shells on the sediment boundary layer undergoes several stages. Rather unexpectedly, even a very small addition of shells (2.5 %) to bare sands provoked a large increase in critical BSS during the initial flat bed stage of the ACC flow experimental runs. This means that much higher turbulence levels were needed to set the sediment in motion, hence the erodibility of the sediment was significantly decreased, by the addition of the small amount of shells to the bare sand. This was followed by a subsequent increase in erodibility towards 15 % shell content, which then exhibited a very slight increase again towards 50 % shell content.

In contrast, the critical depth-averaged velocity required to initiate motion was the largest for bare sands, showing a decrease from 0 to 15 % shell content, followed by a more-gradual decrease from 20 to 50 % shell content. This indicates a small, overall destabilizing effect of shells on sandy sediments.

      The opposing behavior in terms of critical BSS and depth-averaged velocity indicates that shells may modify sediment-flow interactions in two ways: 1) by stabilizing the sediment, and 2) by increasing the effective bed roughness and

near-bed TKE. Following this, the large increase of critical BSS for low shell concentration is probably a consequence of a large increase in sediment stability or by a large increase in bed roughness, given that the reduction in critical depth-averaged velocity remains minimal. In the case of low shell density, shells may disrupt flow in the boundary layer and thereby increase the TKE. For higher shell densities, flow may be deflected over the shells, which progressively reduces the disturbance of the





boundary layer and thus, the TKE. Similar density-dependent alterations in flow pattern from flume studies using either live
animals or mimics have also been observed (Friedrichs et al., 2000, 2009). In these studies, the erosion fluxes and deposition
of suspended material were substantially enhanced when densities were such that less than 4 % of the sediment area was
covered, while above this coverage, both factors saw a drastic reduction.

As the flat bed transitions towards a rippled one, the initial flow and (de)stabilization effects begin to shift. As the
shell content increases, the sand available for migration decreases while the immobile shells hamper ripple formation.
Consequently, the attainable ripple size negatively correlates to shell content. Both the presence of ripples and shells increase
the bottom roughness, and the pattern of the calculated effective bed roughness, which is minimal at intermediate shell content,
shows both impacts. Bed roughness was actually the largest where the shell content was also highest (Figure 8), despite the
ripple size having diminished substantially. This contrasting pattern shows that, in the absence of ripples, small shell
concentrations generate a high effective bed roughness, but this effect is suppressed by the large ripples that are formed under
these conditions at equilibrium. At high shell concentrations however, the direct effect of shells on effective bed roughness is
smaller, but when reinforced by the presence of small ripples, results in a higher combined net roughness (Figure 8).

**4.3 Potential implications of shells for larger-scale sediment dynamics**

Natural sediments rarely consist of pure, clean sand, and often include other debris, fragments and particles (Malarkey et al.,
2015). But sediment characteristics are important for bedform development, roughness and larger-scale implications, and even
minute changes can immediately impact smaller bedforms such as sand ripples. Similar dampening effects have been shown
for other biogenic substances and fine particles (Friend et al., 2008; van Ledden et al., 2004; Malarkey et al., 2015). Biogenic
shells, given their size, density and dimensional aspects, behave very differently from sand grains (Soulsby, 1997), and, as
shown here, a composition of 2.5 % shell can already drastically enhance critical BSS and effective bed roughness. As the
rippled bed matures, which is likely the realistic scenario in many sandy seabeds, the effects of increasing shell content
becomes more evident, through patterns of bed stabilization (e.g., armoring). Our quantities of shell material are well within
the range observed in sandy coastal environments. At a sandy (sand wave) location within the Dutch North Sea (Cheng et al.,
2020; Damveld et al., 2018), the shell content of the sediment samples was also determined. We measured shell percentages
ranging from < 1.0 to 41 % (mean = 8 %, mode = 7 %). Given the observed complexity in the near-bed flow conditions at
these shell percentages, this signifies that many such sandy environments are likely to be subjected to similar sand-shell-ripple
interactions.

The primary mechanisms driving current-generated ripple dynamics are rather well established, but good indicators
are still lacking for ripple size, which is dependent on the grain size, viscosity, density and flow strength (Lapôtre et al., 2017).
Most model predictions typically omit other particle types or represent the sediment by a single value (e.g., $D_{50}$). Sediment
gains and losses due to resuspension or deposition are typically absent (van den Berg, 1987), and attempts have been made to
account for this by including sediment density (van Rijn, 1984, 2006). Ripple size is generally thought to scale with the
thickness of the viscous sublayer (Lapôtre et al., 2017; Yalin, 1985) and does not change with velocity (Baas, 1994). At a





given shields value, which is the nondimensional number that is used to calculate sediment motion, coarser-grained bedforms migrate faster than finer-grained ones (Baas et al., 2000; Lichtman et al., 2018). Yet, this clearly does not hold true for shell

valves and fragments, which cannot be accurately approximated by equations developed for average sand grains. In fact, coarsening due to shell valves and fragments dampened the ripples up to 7-fold with height, more than 2-fold in length and with an order of magnitude reduction in migration rate (bare sand vs. 50 % shells).

We have shown how shell percentages around 10–15 % already reduced ripple size significantly, and above 20 %, ripples are almost entirely absent. We also observed largely-immobile clusters of shells, essentially stabilizing the sediment

through an armoring effect. This is perhaps most comparable with the riverine gravel-bed armoring phenomenon, where due to the coarser sediment particles and flow conditions, coarser grains are partitioned to the top. Consequently, the surface becomes a relatively immobile layer inhibiting sediment transport, among other hydrodynamic interactions (Curran, 2010; Dietrich et al., 1989; Shen and Lu, 1983; Tuijnder et al., 2009). Storm events are often necessary to cause significant flushing of the lower layers or even break an armored layer (Vericat et al., 2006). It would be interesting to investigate how shell clusters

would behave under such extreme conditions. Some evidence suggests that gravel bed armoring can persist through floods, but the level of mobility and partial replacement or renewal of grains in the surface layer is inconclusive (Wilcock and Detemple, 2005).

Care must be taken in drawing comparisons as these are dissimilar environments with entirely different causes for the armoring. Unlike the riverine gravel, which is closer to a spherical shape, shells are an entirely separate class of materials with

biological origins. Moreover, the flow that is characteristic of our study typically consists of diurnal or semidiurnal tides, instead of unidirectional flows. As of yet, it is uncertain how oscillating flow might impose further complexities on a shell-laden sandy bed. However, the relationship between shells and ripples is neither linear or even always positively correlated. Normally, in current-generated ripples, the motion is dominated primarily by flow-induced shear stresses, while immobile materials as shells enhance the turbulence in smoother beds and provide stability in rougher beds. Depending on the bed profile,

shell content can either enhance or reduce sediment motion and ripple migration. Thus, our two types of experiments yield valuable information since measurements on the shifts in boundary layer conditions that occur early on are not visually detectable or quantifiable from the analyses at equilibrium. Under typical unidirectional flow conditions, a higher shell % can be expected to dampen ripple development, migration and, consequently, the bedload transport. How shells might affect the hydrodynamics and bed morphology under more-complex systems and flow conditions, particularly in shallower, wave-

dominated environments, remains to be investigated (e.g., under sheet or oscillatory flow conditions; Nelson et al., 2013; Precht & Huettel, 2003; Soulsby, 1997).

Nevertheless, we foresee many relevant implications of shell research in geomorphologic investigations as well as coastal engineering applications. Shells clearly have the ability to regulate ripple growth and migration, and consequently the bedload transport. A good estimation on the sediment-shell composition would allow us to assess the sediment dynamics for

a given sandy environment and provide better insight on bed stability to produce more-accurate calculations on bedload transport. Concurrently, given the close-coupling between sediment transport and larger-scale adaptations in seabed



morphology, this information could aid in developing or utilizing better methods with regards to offshore seabed patterns, shoreline preservation, longshore sediment transport and coastal management. Our study has provided new insight on how shell material directly, and measurably, influences ripple evolution and migration in fine sand.

## 5 Conclusions

A series of sand-shell-ripple experiments were conducted to directly measure the impact of shell material on the development of current-driven ripples in sandy sediment ($D_{50} = 352\,\mu m$). Our results demonstrate that the shell content has a dynamic effect on the nearbed hydrodynamics that changes over several stages. This mainly occurs as the BSS to flow velocity balance is
430 altered, initially showing a more significant sheltering effect at low shell content ($\leq 15\,\%$) since higher shell quantities will disproportionately enhance the turbulence under a flat bed setting. However, when a sufficient flow velocity is achieved to generate ripples, the shell-induced turbulence will quickly be overcome by the developing bedforms and offset the initial trend. The armoring effect grows stronger with increasing shell content in the form of immobile shell clusters.

In terms of sedimentary transport, shell compositions above 15–20 % exhibit a drastic change in the ability of ripples
to develop and migrate. The threshold is somewhat higher in the constant flow than in the ACC flow experiment (20 % vs. 15 % shells), given the much longer exposure to higher velocity and equilibrium conditions. A sandy mixture with 2.5–50 % shell content increasingly dampens the ripples, thereby reducing the ripple migration by up to one order of magnitude. Moreover, these percentages are representative of certain areas within the natural environment. Thus, the presence of shells needs to be taken into account to better understand and predict the sedimentary processes, as compared to the more simplistic conditions
that could be expected from purely siliciclastic sediment. Our experiments shed some light on the direct influences of shells on ripple dynamics in sandy sediment under unidirectional current-flow conditions. This work would greatly benefit from further studies utilizing other grain sizes combined with shells, as well as an investigation on the other particles of different origin, size, shape and density, but which are nevertheless also commonly found throughout the marine environment.

**Supplementary Materials:** Fig. S1; Table S1; Table S2

**CRediT authorship contribution statement**

**Chiu H. Cheng**: Conceptualization, Methodology, Investigation, Formal analysis, Data Curation, Visualization, Writing – Original Draft, Writing – Review & Editing, Project administration. **Jaco. C. de Smit**: Conceptualization,
Methodology, Investigation, Formal analysis, Software, Visualization, Writing – Original Draft, Writing – Review & Editing. **Greg. S. Fivash**: Methodology, Formal analysis, Software, Visualization, Writing – Original Draft, Writing – Review & Editing. **Suzanne J. M. H. Hulscher:** Writing – Review & Editing, Funding acquisition. **Bas W. Borsje:** Conceptualization, Writing – Review & Editing, Supervision. **Karline Soetaert:** Conceptualization, Writing – Review & Editing, Supervision, Resources, Funding acquisition.





**Data availability.** The data collected and used for this publication will be uploaded to the 4TU.ResearchData repository at the following link:  (10.4121/12852113), hosted by TU Delft, The Netherlands.

**Competing interests.** The author declares that there is no conflict of interest.

**Acknowledgements.** We would like to thank Tjeerd Bouma in the planning and conceptualization of the experiment. Many thanks also to Lennart van IJzerloo, Bert Sinke and Arne den Toonder for their assistance with the setting up of the flume.

**Financial support.** This work is part of the NWO-ALW funded SANDBOX project. The Royal Boskalis Westminster N.V.
and the Royal Netherlands Institute for Sea Research (NIOZ) are also acknowledged for their financial support of this project.

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
