# Peer review of "Grain size characteristics of the sandy sediment fraction used in the experiments."

_Earth Surface Dynamics, 2021_

## Referee Comment (RC2)

This manuscript tackles an interesting subject, the effect of shells on the development and stability of small-scale current ripples. It also investigates the initiation of motion of mixed sand-shell sediment beds. The rationale for the research and the aims and objectives are clear and the methods are described reasonably well. However, the results are not described in sufficient detail to allow me to verify if the interpretations fully support the data. The manuscript lacks what I regard as essential descriptions of the development of the ripples from the initial flat bed, as well as the development of the shell clusters. The development of these clusters contradicts with the statement that the shells were immobile. It is also unclear how the ripple data in Fig. 4 were calculated. Assuming these are mean values, which time period and how many ripples were used to calculate these averages. The tiny histograms are not helpful in this. The authors should include time-series of ripple height and length, so to provide stronger evidence that equilibrium ripples formed at all shell contents. This is important because this would strengthen the argument that the presence of shells leads to smaller equilibrium ripples rather than that the shells delay ripple development because of reduced availability of movable sand due to bed partial bed armouring by the shells, thus resulting in final non-equilibrium ripples of increasingly smaller size as shell content was increased.

Moreover, an explanation of the shape of the TKE curves, and changes in their shape, in Figs 5 and 6 are wanting, and the trends in BBS(cr), u(cr), and ks in Figs 7 and 8 need to be fully explained. I also feel that the authors brush over the fact that their armour layer covers only a small portion of the bed (seemingly much less than half the bed in Fig. 3). Is this really an armour layer in the common meaning of the term? The trend in ripple asymmetry is also not explained.

The relative roles of TKE, bed roughness, and bed stabilisation in bed shear stress trends and ripple size need to be further explored. This would strengthen the auithors' rather speculative conclusions and render their work more applicable. I believe that their data would allow the authors to take this extra step.

I feel that part of the wider implications Section 4.4 is too 'hand-waving', with too many statements that are insufficiently well explored or distract from the main aim of the work (e.g. oscillatory flows)

The writing needs attention throughout, but especially in the Discussion section. The text is also repetitive in a few places.

In summary, this manuscript addresses a novel and timely subject, but it needs further analysis to convincingly show that the interpretations are supported by the data. This may be a matter of describing the data in more detail and strengthening the conclusions.

**Further comments**

Line 62-64 – The ripples described by Baas et al., 2000, Baas & De Koning (21995) and Lichtman et al. (2018) are not around 1 m in wavelength and 0.01 m or more in height, but of the order of 100 mm long and 10 mm high. These are 'current ripples' according to the classification scheme of Ashley et al. (1990, J. Sed. Petrol, v.60, p.160-172), whereas the dimensions given here are in the 'dune' category. This is an important distinction, because current ripples are much more than just "tiny dunes". Their interaction with flows is fundamentally different, reflected in fundamentally different size predictors, for example.

L.68-70 – Lichtman et al. (2018) and Malarkey et al. (2015) did not study "particulate organic matter", but extracellular polymeric substances (EPS), which are cohesive, non-particulate organics.

L.77-78 – What is mean by "general profile"?

L.97-98 – Please rephrase, because most rocks fragments are siliciclastic, so this sentence does not make sense. In which way are shells "hydraulically somewhat more similar to siliciclastic particles"? Please explain.

L.101 – Shells cannot be dead. Replace with "empty shells"?

L.107 – What is meant by "bedform … conditions"? Do you mean bedform stability or perhaps bedform dimensions?

Fig. S1 – What do the colours mean in the graphs?

Equation 4 – Is $ks$ the total bed roughness (based grain friction and form drag)?

L.221 – Please specify "sufficient amount"? This is crucial as, for example, Shields and van Rijn have shown in the past.

L.231-234 – Please be more specific: "strongly controlled" how? "all affected" how? "drastic change" how?

L.235-236 – What is meant by "these ripple parameters"? Which parameters?

L.237 – What is "rather immobile"? Were they immobile or not?

L.238 – How does Fig. 3 show disappearing ripples or ripples migrating around clusters? Move "Figure 3" to end of previous sentence?

L.256-264 – Is it necessary to describe the trends in so much detail? To me, the graphs tell the story sufficiently well. This section can be removed, and the $R^2$-values and p-values can be added to the graphs in Fig. 4.

Figure 4 – How many ripples are the average heights, lengths, asymmetries and migration rates based on for each shell content? When were these parameters measured? At the end of each run? Some of the scatter may be caused by the dynamic nature of rippled beds, with heights and lengths changing all the time even at equilibrium conditions (see Baas, 1994). Ideally, the ripples should be measured at multiple times during equilibrium conditions to reduce data scatter. I would also like to see time-series of ripple height and length: (a) to see when the ripples at low shell content reached equilibrium compared to the control; (b) to check if the ripples at the high shell content were still growing after 4 hours or not. This is important, because the reason for the decrease in ripples height and length could not be related to shells as such, but to the larger size of the shells and fragments (similar to adding gravel to sand).

Figure 5 – The colours of the data points are difficult to distinguish from one another. Please modify.

Figure 5 – Please explain the shape of the TKE curves. And why does the shape change with increasing shell content? I agree that 50% shells has the highest TKE, but is there a trend with increasing shell content from 0 to 50%

L.303-319 – This is a repetition of the results, and therefore unnecessary. In fact, this text reads better (even though it is somewhat convoluted) than the text in the results section. I therefore suggest that the authors move this text to Chapter 3 and merge it with the existing text and focus Chapter 4 on interpretations and discussion.

Section 4.1 – This section does not fully explain the observations and several statements are incomplete or unclear. What kind of structures (L.322)? What is anchoring of shells (L.323)? The text jumps too suddenly from gravel to shells. What does "enhances the erosion threshold" mean (L.323-324), larger or smaller critical shear stress? What is the difference between individual and loose shells (L.325)? If the shells were immobile, how did they form clusters? I am missing a description of how the clusters and armouring layers are formed. Figure 3 does not show an armour layer in the traditional sense, since most of the bed consists of sand. Can the authors rule out that the ripples were small to absent at the highest shell contents because the shells behaved as 'heavy' clasts that were difficult or impossible to move by the flow (thus behaving similar to very coarse sand or gravel clasts)? The smaller ripple sizes may therefore point to progressively slower ripple development rates, with 4 hours being insufficient to form equilibrium ripples at 20% and 50% shells. As mentioned above, this manuscript needs a detailed description of ripple development based on the available video footage. How do the authors explain physically that an enhancement of mean near-bed flow slows down the ripple migration rate? Shouldn't higher velocities increase bed material transport rate and therefore ripple migration rate? The trend in ripple asymmetry in Fig. 4c is not explained.

Section 4.2 – I find the interpretations of the trends in critical bed shear stress and critical velocity too speculative, and there are again unclear statements and comparisons with literature. L.346: Why is this now a gradual decrease, whereas in Section 3 there was no trend? This first paragraph essentially described the data again, which could be considered unnecessary. The second paragraph (L.348-357) raises more questions than it answered to me. Using a simple weighted mean of Shields-derived critical bed shear for 97.5% pure sand ($D_{50}$ = 352 µm; $\tau_{b,cr}$[sand] = 0.2 $Nm^{-2}$) and 2.5% shells ($D_{50}$ = c. 20 mm; $\tau_{b,cr}$[shells] = 18 $Nm^{-2}$) yields $\tau_{b,cr}$[mixed sand-shells] = *c.* 0.7 $Nm^{-2}$, which is close to 0.8 $Nm^{-2}$ in Fig. 7a. Therefore, the increase in critical bed shear stress from 0% to 2.5 % can be explained by the increase in grain size, which is probably similar to what the authors call "stabilising the sediment". Do the authors need to also invoke increasing *ks* and near-bed TKE to explain the increase in critical bed shear stress? In fact, the authors sit on the fence too much, I feel. Would it be possible to do additional analysis and determine the relative contribution of stabilisation, TKE and bed roughness? This would make the paper much stronger, less speculative. Speculation continues in the interpretation of the trends at shell content above 2.5%. What is meant by "deflected over the shells" and how does this reduce the disturbance of the boundary layer? Even if this is a valid explanation, why does the critical bed shear stress first decrease and then increases again. The authors do not explain this. I may be missing something, but didn't Friedrichs et al. find exactly the opposite to this study? L.358-366: This appears to be a convolute way of saying that up to 10% shells the decreasing ripple size leads the falling bed roughness and above 10% shells the increasing shell content leads the rising bed roughness.

L.369-370 – What kind of debris, fragments and particles? Malarkey et al. (2015) did not study debris etc. but EPS, which is non-particulate.

L.388-389 – This statement is only valid for ripples where all particles can be moved. The shells were not moving, so is it surprising that the migration rate was lower? Fewer sand grain were exposed in the mixed sand-shell beds, so the answer here probably is that the ripple migrated slower because less sand was available for bedload transport.

L.403-416 – I do not see what this information adds to the narrative. What is meant by "the flow hat is characteristic of our study typically consists of diurnal or semidiurnal tides, instead of unidirectional flows"? Bidirectional tides are entirely different from unidirectional flows. If his study

really intended to simulate tidal flows, the use of unidirectional flow experiments needs to be fully justified in the introduction and methods. Ripples in oscillatory are also entirely different, so the rather vague statements on wave-dominated environments add nothing substantial to the paper in my opinion. The other statements in this paragraph can be removed or they should be explored/ explained in much more detail, e.g. the potentially interesting sentence on L.410-412.

L.424 – The most commonly used Wentworth scale states that 0.352 mm sand is medium-grained sand.

---

## Author Response (AR2)

Dear Dr. Parsons,

Please find our 2nd submission of the revisions to the manuscript (esurf-2021-13) titled "Sediment shell-content diminishes current-driven sand ripple development and migration" attached for further review and consideration as a research article for *Earth Surface Dynamics*.

To clarify the outstanding points, we have provided additional responses to both reviewers. For Reviewer 1, we specifically elaborated on Point 1, and have added the quadrant analysis figures in the supplementary document, as we feel that they provide an interesting insight about the near-bed turbulence. In addition, we provide responses to the opening remarks of Reviewer 2 (Dr. Jaco Baas).

We provide our responses through an itemized list (original comments in blue text, and comments from this submission in red text) to each of the questions and remarks raised by the reviewers. We have indicated the ways in which we have addressed all the comments in the revised manuscript. All changes in the manuscript itself, as well as the supplementary document, have been done with track changes. For the newest additions to these files, we have additionally bolded the text to make clear what was added in the 2nd submission.

Yours sincerely,

Chiu Cheng and co-authors.
Royal Netherlands Institute for Sea Research

**Subject:**                                                                                          **Date:**
Revisions manuscript ID: esurf-2021-13                                            29 July 2021

Dear Reviewer 1 and Prof. dr. Baas,

Please find attached an updated submission of our revised manuscript titled "Sediment shell-content diminishes current-driven sand ripple development and migration" (manuscript # esurf-2021-13), with minor modifications to the previous submission. We are very pleased with the valuable comments provided, which helped us to further improve the manuscript.

Please find an itemized list of our original responses (in blue text) to each of the questions and remarks raised below. Our new comments below are denoted in red text. We have indicated the ways in which we have addressed all the comments in the revised manuscript. All changes in the manuscript itself, as well as the supplementary document, have been done with track changes. For the newest additions to these files, we have additionally bolded the text to make clear what was added in this 2$^{nd}$ submission.

We hope that all comments and suggestions have been dealt with satisfactorily, and hope that the revised manuscript will be acceptable for publication in the journal, *Earth Surface Dynamics*.

Yours sincerely,
Chiu Cheng, and co-authors

**Comments and Suggestions for Authors**

Reviewer 1

Cheng et al. investigate the affects of shells and shell fragments on the formations and migration of ripples under uni-directional flows. They conducted two sets of experiments to investigate (1) ripple morphodynamics and (2) flow conditions of incipient motion with increasing concentrations of shells/shell fragments. The authors find that increasing shell concentrations drastically impacts ripple morphodynamics as well as flow characteristics needed for incipient motion. Overall, this study is well conducted and yields intriguing results and discussion points. Below is one main comment followed by a few minor comments/questions.

Response to Reviewer 1 Comments
**Point 1:**
How do fluid turbulent structures vary with increasing shell concentrations? I'd be interested to see how near bed fluid velocity fluctuations vary in each of your experiments as you increase shell concentrations. I recommend conducting quadrant analysis (or octant, since you have cross-stream ADV data as well) and seeing if there are any discernible differences as you increase shell concentrations.

**Response 1:** For the cross-shore measurements, we believe that there should ideally be multiple sensors and/or measured positions across the flume width to fully capture the variability in the near-bed velocity/fluid turbulent structures. In our case, there was only one ADV deployed at the same exact position at the center of the flume in each run. Given the conditions of our experimental runs, where the sediment bed was in constant change following incipient motion, we are uncertain whether it would be possible to separate out the effects of turbulent structure from other more-complex form resistance effects (as discussed in Keylock et al. 2014).

However, we have conducted a quadrant analysis on the ADV data for both experiments. In all cases, the turbulence-induced flow appears to be directed forwards and downwards. Particularly for the constant flow experiment, the turbulent structures maintain a similar pattern regardless of the treatment (shell content). However, the size of the turbulence pattern varies between treatments, first decreasing from the control run to 10% shells, then increasing until the 50% run. Notably, there is a drastic reduction in the size of the 15% shell. This may (partially) be due to the fact that the bed was very slightly lower compared to the other runs, while the ADV height remained unchanged over the course of each run. This overall difference between the sensor and bed position appears to have a visible effect exceeding the bed condition itself.

By taking each point of the ADV measurements, we calculate the turbulent velocity component and divide it by the standard deviation of all turbulent velocities over the entire time series of each experimental run. Through these figures, we can clearly see the directional component of the TKE, based on the distribution of the points which are shown by the contours.  What is clear from these figures is that the effect of shell density, whether in the flat bed (acceleration flow experiment) or rippled bed case, on the turbulence is not linearly correlated. This is particularly the case in the ACC experiment, which is also what we already observe in Fig. 5-8, although these do not show the directional component of TKE.  Therefore, we believe that the quadrant analysis figures could be an interesting addition to the paper as a supplemental figure (Fig. S4):

[Figure]

[Figure]

**Figure S4:** The quadrants of each constant flow experimental run, showing the presence of turbulent coherent structures, plotted along the streamwise and vertical planes. The contours represent the point density. Quadrant 2 (top left) represents the burst, or ejection (away from the bed) and quadrant 4 signifies the sweep events in the flow (towards the bed). Exact values are not indicated in the color scale bar as the relative turbulence frequency differs somewhat between each treatment.

**Acceleration flow (flatbed) experiments**

[Figure]

[Figure]

**Figure S5:** The quadrants of each acceleration flow experimental run, showing the presence of turbulent coherent structures, plotted along the streamwise and vertical planes. The contours represent the point density. The burst is represented by Quadrant 2 (top left) and the sweep by Quadrant 4 (bottom right). Exact values are not indicated in the color scale bar as the relative turbulence frequency differs somewhat between each treatment.

Along these same lines, you might also consider calculating the Reynold's stress for comparison to your calculations of bottom shear stress. References to consider:

- Keylock, C. J., S. N. Lane, and K. S. Richards (2014), Quadrant/octant sequencing and the role of coherent structures in bed load sediment entrainment, J. Geophys. Res. Earth Surf., 119, 264–286, doi:10.1002/ 2012JF002698.
- Bogard, D. G., and W. G. Tiederman (1986), Burst detection with single-point velocity measurements, J. Fluid Mech., 162, 389–413

We calculated the near-bed fluid velocity fluctuations through the TKE, which is practically proportional to the Reynold's stress. This is a more robust method than using e.g. quadrant analysis or Reynolds' stress, as these are highly sensitive to the orientation of the ADV.

We have specified this in the text (lines 194-196):

"The near-bed turbulent kinetic energy (TKE) was derived from the near-bed flow velocity fluctuations (Pope et al., 2006). This value indicates the mean kinetic energy associated with eddies from the turbulent flow. It is a more-robust method for determining the bed shear stress than e.g., quadrant analysis or Reynold's stress, as these are highly sensitive to the orientation of the ADV profiler."

And with reference to the new supplementary quadrant analysis figures:

"Overall, there is also a consistent pattern in the turbulent structure maintained between each run (Fig. S4)." (Lines 304-305)

"The quadrant analysis plots show that the turbulence-induced flow is predominantly directed forwards and downwards (Fig. S5)." (Lines 321-322)

**Point 2:** Are the shells ever incorporated into the ripples or do they primarily armor the bed and the ripples migrate over them? If the latter, could the armoring essentially restrict the sediment supply for the ripples in that they can no longer entrain additional sediment from below them?

**Response 2:** What we visually observe is that ripples frequently migrated over the shells at the lower percentages (below 15 – 20%). As shell content increased and ripples decreased in size, migration over the shells diminished accordingly. Especially in the 40% and 50% shell treatments, the small ripples sometimes migrated around the lower parts of the shell cluster, or ceased entirely.

In all shell-containing treatments, the majority of the shells were not moved by the migrating ripples. In the 5 % experiment, a loose surficial shell would occasionally be observed to shift a few mm due to flow or a migrating ripple, but would remain in place (sometimes a change in orientation). This is most clearly seen from the GoPro footage of the constant flow experiments, which can be found at: https://figshare.com/s/a9edfa562c7fde7ef95f

Therefore, the armoring appears to restrict further sediment entrainment beneath the shells, although we did not measure this quantitatively.

**Author's changes:** (Lines 255-257)
"These bands of shells were  immobile, and the already-smaller ripples were observed from the GoPro videos to either migrate around the  denser and slightly highe-positioned shells, or disappear altogether, so the shells did not incorporate themselves into the (migrating) ripples. Even in the lower shell concentrations, where larger ripples frequently migrated over the sparser quantity of shells, the vast majority of these surficial shells were not moved by either the moving ripples or flow (Fig. S2)."

The two figures below show 16 snapshots extracted from GoPro footage of the 5 and 50% runs (constant flow experiment), and have been added as Supplementary Fig. S2 to support the statement above, with also a short statement about shell immobility.

**5 % shell content (numbers = minutes)**

[Figure]

**50 % shell content (numbers = minutes)**

[Figure]

**Figure S2**: Snapshots extracted from the GoPro footage of the constant flow experiment to show that the majority of the shells remain in place throughout the duration of the experimental run at both the low and high shell treatments.

**Point 3:** Throughout the manuscript you refer to near-bed flow in the "horizontal" direction. I think "streamwise" would be a better word to use as "horizontal" could apply to either the cross-stream (y) or streamwise (x) directions.

**Response 3:** We agree with your suggestion and now refer to all instances of near-bed horizontal flow as "streamwise" (Lines 301 and 324).

**Point 4:** Any time-averaged variable should have an overbar to denote the time-averaging. I noticed this mainly in figures 5 and 6 but should be applied throughout the paper.

**Response 4:** We have added overbars to Fig. 5 a/b and 6 a/b. Furthermore, we have removed panel C from Fig. 6, per the comments from reviewer 2 about the variability observed in the TKE over depth, which is an artifact. Moreover, this information is more-clearly illustrated through Fig. 7 (BSS) and 8 (effective bed roughness). The figures and captions are revised accordingly.

**Author's changes:** (Lines 306 – 309; 323 - 326)

[Figure]

**Figure 5:** Time-averaged near-bed velocity profiles showing the *(a)* x and *(b)* z direction of the constant flow experimental runs, as well as the *(c)*  peak TKE  values plotted against shell content. *Note*: The profiles are time-averaged, as indicated by the overbars, over the entire duration of each experimental run.

[Figure]

**Figure 6:** *(a)* Near-bed  streamwise flow and, *(b)* vertical flow  at the onset of sediment transport for flat beds (ACC flow experiment). *Note*: The overbars denote that the x-axes  are time-averaged, over a 10-minute period, which encompasses the 5 minutes prior to and following the incipient motion, for the four selected experimental runs.

**Point 5:** In figure 7, what are the black data points? Add a label directly to the figure to denote what these are. You can also add a label directly to the figure for the 95-percentile shaded areas. I tend to lean towards labeling as much as I can in the figure itself rather than "hiding" that information in the captions. Makes it easier for readers to get everything out of the figure without having to flip back and forth with the text.

**Response 5:** The black points were included to denote those treatments as the control (0 % shells). Since they serve no other purpose, we have removed them to avoid any confusion. We have now also labeled the 95-percentile directly on the figure, while also still keeping it in the caption.

**Author's changes:** (Line 327)

[Figure]

**Point 6:** I think it would be worth while to add a table to supplement that summarizes your experimental conditions (essentially a table of the paragraph that starts on line 142).

**Response 6:** We have provided a new table for the supplementary document to summarize the most important experimental aspects. From this, readers can quickly ascertain the settings utilized in each experiment. It is referenced in the manuscript at Line 155.

**Table S3**

Experimental settings and measurements undertaken in both experiments.

| Experiment | $D_{50}$† | Flow (cm s$^{-1}$) | Shell % | Duration | Measurements |
|---|---|---|---|---|---|
| ACC | 352 µm | 15 to 50 (increase of 0.3 min$^{-1}$) | 0, 2.5, 7.5, 10, 12.5, 15, 20, 25, 30, 40, 50 | ~4 hr. 26 min. | ADV†† GoPro Time-lapse‡ |
| CF | 352 µm | 50 | 0, 5, 10, 15, 20, 50 | ~4 hr. 26 min. | ADV††† GoPro‡ Time-lapse |

†Bare sandy sediment (control)
††Average over the 5 minutes before and after incipient motion
†††Average over the entire experiment
‡Data available but not used

**Comments and Suggestions for Authors**

Reviewer 2

This manuscript tackles an interesting subject, the effect of shells on the development and stability of small-scale current ripples. It also investigates the initiation of motion of mixed sand-shell sediment beds. The rationale for the research and the aims and objectives are clear and the methods are described reasonably well.

However, the results are not described in sufficient detail to allow me to verify if the interpretations fully support the data. The manuscript lacks what I regard as essential descriptions of the development of the ripples from the initial flat bed, as well as the development of the shell clusters. The development of these clusters contradicts with the statement that the shells were immobile.

With regards to ripple development, we have clarified the description about incipient motion to better illustrate the transition from a static flatbed towards initial sediment movement. We have also revised the text in instances where reference was made to the "development" of shell clusters. What visually appear as clusters is actually an artefact from the gradual movement of sand, which periodically exposes or buries the most surficial shells. We completely homogenized the sediment mixture in every experimental run, and the shells were immobile throughout the experimental duration at every concentration. Any appearance of shell bands is simply a consequence of surficial shells at different sections of the channel being exposed by ripple migration at differing times. Thus, the shell armouring was there from the start, rather than forming over the course of the experimental runs.

It is also unclear how the ripple data in Fig. 4 were calculated. Assuming these are mean values, which time period and how many ripples were used to calculate these averages. The tiny histograms are not helpful in this. We have provided additional details in the text on how the ripple parameters were calculated, specifically the time period and the approximate number of ripples in each run. The entire experimental duration was used in all of the calculations. However, whereas the ripple height, length and asymmetry were determined from each of the 1600 frames, the migration rate was done over 24-frame intervals. The approximate number of ripples from each run is illustrated by raster plots (new supplemental Fig. S3). See Point 15 below for further details.

The authors should include time-series of ripple height and length, so to provide stronger evidence that equilibrium ripples formed at all shell contents. This is important because this would strengthen the argument that the presence of shells leads to smaller equilibrium ripples rather than that the shells delay ripple development because of reduced availability of movable sand due to bed partial bed armouring by the shells, thus resulting in final non-equilibrium ripples of increasingly smaller size as shell content was increased.

For ripple development, we have produced time-series plots for the height, length and migration rate to show that the ripples reached equilibrium within about the first hour of the experiment (also part of Point 15). We are confident that our experimental runs illustrated scenarios in which increasing shell content lead to smaller (equilibrium) ripples, rather than simply delaying the ripple development. An increase in the shell content decreases the quantity of moveable sand, and this is also a cause for the reduction in ripple size.

Moreover, an explanation of the shape of the TKE curves, and changes in their shape, in Figs 5 and 6 are wanting, and the trends in BBS(cr), u(cr), and ks in Figs 7 and 8 need to be fully explained. Figure 5c has been modified to plot the peak TKE values against the shell content to show the trend much more clearly than the TKE profiles over the measurement depth. An increasing trend is not observed from 0 to 50%. The peak value first decreases from 0 to 10% shells, after which there is an increase up to 50% shells. The shape of the curve has to do with the proximity of the ADV to the bed, as this affects the data quality (e.g., signal noise). The TKE profiles from figure 6c have been removed altogether, as the BSS information from Figure 7a provides a clearer illustration of the trend (Point 17).

Since it may not be immediate obvious to the reader, we also point out that Figure 7 refers specifically to the ACC experiment where, under flat bed conditions, a small increase in shells enhances bed roughness because it is no longer completely smooth. But since the shells are immobile even at the lower concentrations, there is an enhancement in bed stability even though the critical velocity in Figure 7b shows that the shells produce a small destabilizing effect. This would explain why the total bed roughness initially increases in Figure 8 for the ACC experiment, but quickly drops beyond 7.5% shell content, apparently due to the density-driven effect by the shells. In practice, the stabilizing effect increases with the shell content, as reflected by the reduction in ripple size in the constant flow experiment. Here, the total bed roughness initially decreases from 0 to 10% shell content, but then increases and even exceeds the control situation at 50%. However, the reduction in ripple size is consistent from 0 to 50%, because of the reduction in available sediment and bed stabilization by the immobile shells.

I also feel that the authors brush over the fact that their armour layer covers only a small portion of the bed (seemingly much less than half the bed in Fig. 3). Is this really an armour layer in the common meaning of the term? The trend in ripple asymmetry is also not explained.

The surficial coverage of this "armor layer" is not static, since there is periodic movement of sand ripples across the flume channel following incipient motion. At any given time, the visible coverage may appear to be less than half of the total area, but the surficial shells are periodically, often repeatedly, exposed or buried over most of the visible surface area. A few snapshots do not fully convey the extent of the shell density/distribution. Supplementary figure S2 provides an improved representation of areal shell coverage, although the primary purpose of this figure is to highlight the immobility of the surficial shells over the experimental duration. We believe that the GoPro videos most clearly illustrate the shell distribution information, and have made these available on the public repository (https://figshare.com/s/a9edfa562c7fde7ef95f; also see Point 15).

The relative roles of TKE, bed roughness, and bed stabilisation in bed shear stress trends and ripple size need to be further explored. This would strengthen the auithors' rather speculative conclusions and render their work more applicable. I believe that their data would allow the authors to take this extra step.

As suggested by Reviewer 1, we have also conducted a quadrant analysis to illustrate the near-bed turbulent structures (supplementary Fig. S4 and S5) to show, in essence, the directional component of the TKE. These figures show that the overall turbulent structures remain mostly consistent between each run in both experiments. Most of the flow was directed downwards and forwards, although under the 0% flat bed condition the size of the turbulent pattern is much reduced. The bed condition here is the smoothest. We hope that these, combined with the time-series and raster plots, present a clearer picture and more-thorough analysis on the effect of the shell content on ripple development and near-bed flow.

I feel that part of the wider implications Section 4.4 is too 'hand-waving', with too many statements that are insufficiently well explored or distract from the main aim of the work (e.g. oscillatory flows)

We have largely deleted the speculative statements in this subsection (see Points 20 and 23). In particular, a large part of paragraphs 2 and 4 from the of the final subsection of the discussion (now 4.2) have been removed as they were not specifically tested in our study and/or do not add to the relevance of our findings. For example, the reference to tidal oscillation and wave-generated ripples have been removed.

The writing needs attention throughout, but especially in the Discussion section. The text is also repetitive in a few places.

The text has been trimmed in several of the sections. The results from the ripple parameters have been reduced, with some of the deleted information put into Figure 4 directly. The opening paragraphs of the discussion, as well as paragraph 1 of subsection 4.2 were a partial repetition of the

results/general observations. Some of the relevant sentences were shifted into the results, while the rest was deleted (Points 18 and 20).

In summary, this manuscript addresses a novel and timely subject, but it needs further analysis to convincingly show that the interpretations are supported by the data. This may be a matter of describing the data in more detail and strengthening the conclusions.

Response to Reviewer 2 Comments

**Point 1:**
Line 62-64 – The ripples described by Baas et al., 2000, Baas & De Koning (21995) and Lichtman et al. (2018) are not around 1 m in wavelength and 0.01 m or more in height, but of the order of 100 mm long and 10 mm high. These are 'current ripples' according to the classification scheme of Ashley et al. (1990, J. Sed. Petrol, v.60, p.160-172), whereas the dimensions given here are in the 'dune' category. This is an important distinction, because current ripples are much more than just "tiny dunes". Their interaction with flows is fundamentally different, reflected in fundamentally different size predictors, for example.

**Response 1:** Thank you for this point of clarification. We are mainly interested in the 'current ripples' as defined in Ashley et al. (1990), and have made the change accordingly to the text. We also removed two references not dealing specifically with these ripples and added Ashley et al. (1990).

**Author's changes:** (Lines 63-66)
"…with typical sizes of around 0.1 m in wavelength and up to 0.01 m or more in height (Ashley et al., 1990; van Rijn et al., 1993) (Knaapen et al. 2005; van Rijn et al., 1993). They continuously develop and erode, typically on the order of minutes to days, and can migrate at rates exceeding 0.4 cm min$^{-1}$ (Baas et al., 2000; Baas and De Koning, 1995; Bartholdy et al., 2015; Lichtman et al., 2018; Miles et al., 2014)"

**Point 2:**
L.68-70 – Lichtman et al. (2018) and Malarkey et al. (2015) did not study "particulate organic matter", but extracellular polymeric substances (EPS), which are cohesive, non-particulate organics.
**Response 2:** We have removed these two references, as we do not want to introduce potential confounding factors that affect sediment stability/erodibility from non-particulate, surficial binding organic materials like EPS or biofilms in general. (Line 71)

**Point 3:**
L.77-78 – What is mean by "general profile"?
**Response 3:** We meant to describe the "general composition" of the sediment, and have replaced "profile" with "composition." (Line 80)

**Point 4:**
L.97-98 – Please rephrase, because most rocks fragments are siliciclastic, so this sentence does not make sense. In which way are shells "hydraulically somewhat more similar to siliciclastic particles"? Please explain.
**Response 4:** In the study by Al-Dabbas and McManus, they explored the possibility of using *Mytilus edulis* (blue mussel) shell debris as a tracer, with the inference that it could be a potential proxy for finer sand grain motion within an estuary setting, specifically the Tay Estuary, UK. They mention that shells exhibit hydraulic similarity to smaller sand particles, and that this has been repeatedly shown in flume studies. However, they do not cite any of the studies. Therefore, we have reworded the first two sentences of the paragraph.

**Author's changes:** (Lines 99-103)
"On the one hand, shells behave differently than rock and other inorganic fragments of similar size in that they are hydraulically somewhat more similar to siliciclastic particles, even when the sizes differ greatlyFrom a hydraulic point of view, biogenic materials such as shells do not exhibit the same response as compared to rock fragments of a similar size, although some shells (e.g., the mussel family, Mytilidae) have been shown to behave more similarly to the smaller sand particles (Al-Dabbas and McManus, 1987). On the other handHowever, due to the shape and size of most shells….."

**Point 5:**
L.101 – Shells cannot be dead. Replace with "empty shells"?
**Response 5:** We agree that the suggested term would be clearer and have made the change. (Lines 105 and 118-119)

**Point 6:**
L.107 – What is meant by "bedform … conditions"? Do you mean bedform stability or perhaps bedform dimensions?
**Response 6:** We meant the stability of the sediment bed, and have modified the sentence:

**Author's changes:** (Lines 111-112)
"…largely dictates the sediment dynamics, thereby affecting bedform development and  sediment stability…"

**Point 7:**
Fig. S1 – What do the colours mean in the graphs?
**Response 7:** Both panels (a) and (b) from Fig. S1 show all of the data points together (whole shells and fragments) to put them into perspective of one another. Vibrant colors highlight the whole shell measurements in Fig. S1a, and the fragment measurements in Fig. S1b. Furthermore, the colors indicate the point density, based on a kernel density estimate, where red indicates length – height ratios which are very common while blue indicates rare length – height ratios. In addition, we noticed that the x-axis label was incorrect. It should be "Length" because "Diameter" and "Height" are actually the same parameter and "thickness" belongs to Height and not the Length:

**Author's changes:** (Supplementary files.docx)

[Figure]

"**Figure S1**: Measured shell Height (thickness) vs. Length  for (a) whole shell valves and (b) shell fragments. Both panels contain all of the measurements, but the dataset of interest is indicated by the bright colors, while the points from the other dataset are plotted in grayscale. Colors indicate point density (e.g., how often a specific length – height ratio occurs with respect to the whole dataset) based on a kernel density estimate."

**Point 8:**
Equation 4 – Is ks the total bed roughness (based grain friction and form drag)?
**Response 8:** Given that we back-calculate this from the TKE measurements, it is indeed the total bed roughness. We have changed all references of "effective bed roughness" to "total bed roughness" and with the following addition:

**Author's changes:** (Lines 220-221)
"… and ks is the total bed roughness (m) by combined grain friction and form drag."

**Point 9:**
L.221 – Please specify "sufficient amount"? This is crucial as, for example, Shields and van Rijn have shown in the past.
**Response 9:** This is the point at which we visually observe frequent and constant motion of sand grain across the entire flume area. The Shields curve represents conditions with substantial movement of particles (frequent particle movement at all locations; about 10%–50% of bed is moving; van Rijn 1993). We have specified this in the text:

**Author's changes:** (Lines 233-234)
"The onset of incipient motion, which was defined as the frequent movement of particles across the entire flume area , was derived visually from the GoPro footage (van Rijn 1993)."

**Point 10:**
L.231-234 – Please be more specific: "strongly controlled" how? "all affected" how? "drastic change" how?
**Response 10:** We have clarified the statements through the following changes:

**Author's changes:** (Lines 244-248)
"…the results clearly demonstrate that the  reduction of ripples is strongly  correlated  to the shell fraction of sandy sediments. Consequently, the ripple height, length  and migration rate were all  significantly reduced by the increasing shell content, while the ripple shape became slightly more asymmetric.  In the constant flow experiments, the ripples appeared to achieve equilibrium conditions within the first hour at a flow rate of 50 cm s$^{-1}$. The  change in ripple  length and height, in particular, can clearly be seen in the concluding frames…"

**Point 11:**
L.235-236 – What is meant by "these ripple parameters"? Which parameters?
**Response 11:** These are the ripple length, height, asymmetry and migration rate. They have been added to the text, with clarification on why they were not measured in the ACC experiment.

**Author's changes:** (Lines 249-252)
" The ripple  height, length, asymmetry and migration rate were not measured in the ACC flow experiment as we were interested in determining the incipient motion from these runs. Nevertheless, a similar observation could still be seen at around 15 % shell content, even though these ripples were less equilibrated given the lower flow rates for much of the experimental duration (Fig. 2b)."

**Point 12:**
L.237 – What is "rather immobile"? Were they immobile or not?
**Response 12:** Yes, these shell clusters were immobile. We have removed "rather" to avoid confusion. (Line 254)

**Point 13:**
L.238 – How does Fig. 3 show disappearing ripples or ripples migrating around clusters? Move "Figure 3" to end of previous sentence?
**Response 13:** We have moved the reference to Fig. 3 as suggested (Line 253). That statement better describes the message we intended to convey with that figure.

**Point 14:**
L.256-264 – Is it necessary to describe the trends in so much detail? To me, the graphs tell the story sufficiently well. This section can be removed, and the R2 -values and p-values can be added to the graphs in Fig. 4.
**Response 14:** Thank you for this suggestion to make the results more concise. The statistical information has been moved to the figures. We have cut out most of the description, keeping only the rates for each parameter and merging that with the previous paragraph. (Lines 277-281)

**Author's changes:** (Line 283)

[Figure]

**Point 15:**
Figure 4 – How many ripples are the average heights, lengths, asymmetries and migration rates based on for each shell content?
**Response 15:** Not every treatment contained the same number of ripples. The exact number could vary slightly, depending on how that is determined. Occasionally, two ripples would merge into one, or one ripple might split into two. Sometimes the process would reverse. Based on the time series, we estimate approximately 18, 20, 14, 13, 13 and 12 ripples included in the calculations over the course of each experimental run (from 0 to 50 % shells in the constant flow experiment). Here are raster plots of sediment height over the experimental duration of each treatment. Each ripple is visible from the contrasting extremities in the sediment height. The vertical axis shows the distance that the ripples had traveling along the flume channel over time. This has also been added as supplemental Fig. S3.

[Figure]

**Figure S3**: Raster plots of the sediment height of each constant flow experimental run to show the distance (y-axis) each ripple traveled along the flume channel over time (x-axis). These plots also show the approximate number of ripples that was present and used in the calculation of ripple height, length, asymmetry and migration rate for each run.

When were these parameters measured? At the end of each run? Some of the scatter may be caused by the dynamic nature of rippled beds, with heights and lengths changing all the time even at equilibrium conditions (see Baas, 1994). Ideally, the ripples should be measured at multiple times during equilibrium conditions to reduce data scatter.

The four ripple parameters were not measured at the end of each run, but throughout the entire duration (1600 frames, at 10-second intervals). For the height, length and the asymmetry, each individual frame was a calculation. However, only ripples that were completely within the photo frame were counted. Any ripple that had not yet fully migrated into the frame in the upstream and ripples that had partially moved out of the frame in the downstream were excluded. For the migration rate, this was based on the distance travelled over a 4-minute interval (24 frames), using again only whole ripples. The first calculation is therefore only available from the 24th frame in the analysis, and no calculation is derived from the very last 23 frames in the experimental run. Thus, ripples were measured repeatedly over the entire experiment. This has been added to the methods:

**Author's changes:** (Lines 187-190)
"This frame interval allowed ripples to travel measurable distances while limiting the likelihood of them moving out of frame before measurements could be taken. All four of the ripple parameters were measured throughout the experimental duration using each frame. Only whole ripples were used in the analyses, as ripples that were partially in (upstream) or out (downstream) of the frame were excluded. Given that the migration rate was calculated over 24 frames, measurements were not generated from the first or last 23 frames in each run..."

I would also like to see time-series of ripple height and length: (a) to see when the ripples at low shell content reached equilibrium compared to the control; (b) to check if the ripples at the high shell content were still growing after 4 hours or not. This is important, because the reason for the decrease in ripples height and length could not be related to shells as such, but to the larger size of the shells and fragments (similar to adding gravel to sand).

Here we provide time-series plots for ripple height, length and migration rate of all six treatments. We believe that the ripples reach equilibrium within the first hour, and remain consistent for the remainder of the experiment. The time series plots show all 1600 frames. To clarify the ripple development, we have noted in the first paragraph of the results about ripples achieving equilibrium within the first hour, based on the time-series figures.

[Figure]

**Point 16:**
Figure 5 – The colours of the data points are difficult to distinguish from one another. Please modify.
**Response 16:** We have removed the center black dots from the data points, which was an artifact of the plotting output from Matlab. See Point 17 for new versions and further details about changes, which include removal of subplot "c" from Fig. 6.

**Point 17:**
Figure 5 – Please explain the shape of the TKE curves. And why does the shape change with increasing shell content? I agree that 50% shells has the highest TKE, but is there a trend with increasing shell content from 0 to 50%
**Response 17:**
The shape of the TKE curve does not represent an actual dependence of TKE on elevation above the bed. It is in fact related to the data quality, which depends on the proximity of the bed (which causes reflections that pollute the ADV signal and thereby increase noise. To represent the TKE more

clearly, we now plot the peak TKE values against shell content in subplot 5c. The TKE initially drops under equilibrium ripple conditions, as the ripples are somewhat diminished at the lower shell contents, up to about 10%. Then, it begins to increase as the ripples continue to diminish while the growing shell content increasingly adds to the overall bed roughness.

However, we have removed subplot 6c, since this information is more-clearly illustrated through Fig. 7 (BSS) and 8 (effective bed roughness). Overbars have also been added to show that these are time-averaged variables.

**Author's changes:** (Lines 306 – 309; 323 - 326)

[Figure]

**Figure 5:** Time-averaged near-bed velocity profiles showing the *(a)* x and *(b)* z direction of the constant flow experimental runs, as well as the *(c)*  peak TKE  values plotted against shell content. *Note*: The profiles are time-averaged, as indicated by the overbars, over the entire duration of each experimental run.

[Figure]

**Figure 6:** *(a)* Near-bed  streamwise flow  *(b)* vertical flow  at the onset of sediment transport for flat beds (ACC flow experiment). *Note*: The overbars denote that the x-axes  are time-averaged, over a 10-minute period, which encompasses the 5 minutes prior to and following the incipient motion, for the four selected experimental runs.

**Point 18:**
L.303-319 – This is a repetition of the results, and therefore unnecessary. In fact, this text reads better (even though it is somewhat convoluted) than the text in the results section. I therefore suggest that the authors move this text to Chapter 3 and merge it with the existing text and focus Chapter 4 on interpretations and discussion.

**Response 18:** The latter part of the first paragraph has been integrated into the first paragraph of the results (Lines 259-262), while the rest has been deleted. Most of the 2nd paragraph has been moved to the first paragraph of subsection 3.1 (Line 274-276). A small part of the third paragraph was moved to subsection 3.2 (Lines 314-315), and the rest was deleted.

**Point 19:**
Section 4.1 – This section does not fully explain the observations and several statements are incomplete or unclear. What kind of structures (L.322)? What is anchoring of shells (L.323)? The text jumps too suddenly from gravel to shells. What does "enhances the erosion threshold" mean (L.323- 324), larger or smaller critical shear stress? What is the difference between individual and loose shells (L.325)?

**Response 19:** By "flow" structures, we meant protrusions/topographic roughness on the bed surface. With "anchoring of shells" we meant that the shell valves are fully or partially buried and therefore fixed in the sediment (immobile); this could also be due to being interlocked by the surrounding shells. The critical shear stress is elevated as a result of this, not lowered. By "individual or loose shells", we were referring to shells not buried into the sediment in any way. These points are expressed through the following changes:

**Author's changes:** (Lines 362-366)
"…In gravel bed rivers, it is known that the incorporation of  topography into the sediment surface creates microclusters that increase both the bed roughness as well as bed stability (Curran, 2010). The anchoring of shells, even through partial burial, in sandy sediment greatly  raises their critical erosion threshold compared to individual shells situated on a flat surface, irrespective of the orientation. Whereas  loose shells on top of a flat sandy surface can erode at velocities…"

If the shells were immobile, how did they form clusters? I am missing a description of how the clusters and armouring layers are formed. Figure 3 does not show an armour layer in the traditional sense, since most of the bed consists of sand.

Both the GoPro footage and photo images show that, what would visually appear as shell clusters, the bands of shells were already there from the start, rather than having been formed as a result of the flow and ripple generation. The shells remained completely immobile throughout the entire experiment, especially at the higher shell percentages. Before each experiment, the sediment was thoroughly homogenized. Thus, these visual clusters are really just an artifact from the local presence and movement of sand and are not localized clusters of shells due to heterogeneity in the sediment mixture. We included this information in the discussion.

**Author's changes:** (Lines 367-370)
"In our experiments, the shells were almost completely immobile over the entire duration of the experimental runs, with visually no noticeable change as evidence by both the time series photos and video footage (Fig. S2). What would appear as bands of shell is an artifact caused by the localized changes and movement of the sand, rather than a change to the shells."

Can the authors rule out that the ripples were small to absent at the highest shell contents because the shells behaved as 'heavy' clasts that were difficult or impossible to move by the flow (thus behaving similar to very coarse sand or gravel clasts)?

Indeed, we believe that the immobility of the shells is likely a major cause for the diminished ripple sizes with increasing shell content. We added a short statement about the clusters, in which we explain that this is merely a visual phenomenon as the sand is gradually migrating across the flume channel, occasionally exposing or covering up the surface shells. The shells themselves have been homogenized throughout the sediment mixture and are immobile from the very start of the experiment.

The smaller ripple sizes may therefore point to progressively slower ripple development rates, with 4 hours being insufficient to form equilibrium ripples at 20% and 50% shells. As mentioned above, this manuscript needs a detailed description of ripple development based on the available video footage.

From our time series plots (see Point 15), the ripples do not continue to develop at slower and slower rates over the course of the experiment, but appear to achieve equilibrium within an hour. In fact, during the experiments we visually observed that the ripples reached their equilibrium dimension more quickly, due to them being smaller.

We have GoPro footage to illustrate the large contrasts in ripple size/migration between high and low shell content. While some of the shells would occasionally be covered by migrating ripples, the vast majority of the shells in the upper layers remained immobile throughout the experiment. In the 5 % experiment, a loose surficial shell would occasionally be observed to shift a few mm due to the flow or migrating ripple, but would remain more or less in place. The video footage shows this most clearly, and these can be found on the repository: https://figshare.com/s/a9edfa562c7fde7ef95f

In addition, we have selected 16 sequential snapshots from the video footage for the 5 and 50 % experiments to illustrate how the shells stay in place even after several hours. These figures have been added to the supplemental material:

**5 % shell content (numbers = minutes)**

[Figure]

**50 % shell content (numbers = minutes)**

[Figure]

**Figure S2**: Snapshots extracted from the GoPro footage of the constant flow experiment to show that the majority of the shells remain in place throughout the duration of experimental run at both the low and high shell treatments.

How do the authors explain physically that an enhancement of mean nearbed flow slows down the ripple migration rate? Shouldn't higher velocities increase bed material transport rate and therefore ripple migration rate?

The elevation in the near-bed flow would be expected to enhance ripple migration, and this elevation is most evident at the low percentages of shell, especially under flat bed conditions (ACC experiment). However, we also see that the shells, even at the low concentrations, are largely immobile even when ripples are migrating over them. This stability, combined with an overall reduction in sand availability from shell displacement, causes the ripples to diminish in size in all cases where shells are present. Thus there appears to be a counteraction between enhanced near-bed flow and enhanced bed stability, both induced by the presence of shells, and in which the stability effect outweighs the enhanced flow.

The trend in ripple asymmetry in Fig. 4c is not explained.

Although the asymmetry somewhat increased with shell content, we believe this to be more noise than an actual trend and added a short section stating this. It is the least significant out of the 4 parameters. (Lines 440-441)

**Point 20:**
Section 4.2 – I find the interpretations of the trends in critical bed shear stress and critical velocity too speculative, and there are again unclear statements and comparisons with literature. L.346: Why is this now a gradual decrease, whereas in Section 3 there was no trend? This first paragraph essentially described the data again, which could be considered unnecessary. The second paragraph (L.348-357) raises more questions than it answered to me.

Yes, it should have been no decrease, as stated in Section 3. However, we have deleted the entire first paragraph as that is indeed a repetition of the results (Lines 382 - 392). In addition, part of paragraph 2 and all of paragraph 3 are now combined into Section 4.1

Using a simple weighted mean of Shields derived critical bed shear for 97.5% pure sand (D50 = 352 mm; tb,cr[sand] = 0.2 Nm-2 ) and 2.5% shells (D50 = c. 20 mm; tb,cr[shells] = 18 Nm-2 ) yields tb,cr[mixed sand-shells] = c. 0.7 Nm-2 , which is close to 0.8 Nm-2 in Fig. 7a. Therefore, the increase in critical bed shear stress from 0% to 2.5 % can be explained by the increase in grain size, which is probably similar to what the authors call "stabilising the sediment".

To clarify, we determined the Critical BSS of the sand (within an immobile matrix of shells), not of the entire sediment matrix.

Do the authors need to also invoke increasing ks and near-bed TKE to explain the increase in critical bed shear stress? In fact, the authors sit on the fence too much, I feel. Would it be possible to do additional analysis and determine the relative contribution of stabilisation, TKE and bed roughness? This would make the paper much stronger, less speculative.

We have performed Quadrant analysis on our ADV data, as suggested by the other reviewer (see Point 1 of Reviewer 1), to try and identify patterns in the near-bed turbulent structures due to increasingly shell content. Interestingly, the turbulence patterns are very similar between both experiments and each run therein. It is also directed forwards and downwards, presumably due to the presence of ripples. However, it is also clear that the exact relationship between shell content and the near-bed flow is not very straightforward, as we show in Fig. 7 and 8. As a result, we include these quadrant analysis figures as supplemental figures 4 and 5.

As far as the additional analyses on the relative contribution bed stabilization, TKE and bed roughness, we only measured flow velocity/turbulence and do not feel that there is sufficient information to determine whether the high turbulence is due to increased CBSS or an artifact of increased roughness.

Speculation continues in the interpretation of the trends at shell content above 2.5%. What is meant by "deflected over the shells" and how does this reduce the disturbance of the boundary layer? Even if this is a valid explanation, why does the critical bed shear stress first decrease and then increases again. The authors do not explain this.

**Response 20:** We believe that this is due to the density of the shells. Studies in the past, using a variety of hard structure-mimics in flume studies, have shown that low densities of objects act as turbulent-enhancing roughness structures in the near-bed water column. However, at a certain threshold point, the density actual attenuates the flow around and above the dense patch of protrusions, thereby reducing the turbulence and disturbance to the boundary layer. This is likely also the cause for the initial decrease in critical BSS, followed by a very gradual increase.

I may be missing something, but didn't Friedrichs et al. find exactly the opposite to this study? L.358-366: This appears to be a convolute way of saying that up to 10% shells the decreasing ripple size leads the falling bed roughness and above 10% shells the increasing shell content leads the rising bed roughness.

We try to draw parallels to the Friedrichs et al. studies, where they observed enhanced scouring at low densities of hard structures, relative to no structures, and a density threshold, above which the flow is attenuated and there is less scouring compared to the control situation. Thus, there appears to be a similar, density-driven effect of the shells on the near-bed flow. Similar observations have also been found in both rigid and semi-rigid vegetation structures, including corresponding flume studies that use hard objects as mimics.

**Point 21:**
L.369-370 – What kind of debris, fragments and particles? Malarkey et al. (2015) did not study debris etc. but EPS, which is non-particulate.

**Response 21:** These include both biogenic and inorganic substances that are often found in marine sediments, particularly in close proximity to the coast. We have updated the citation with three new references and removed Malarkey et al. 2015. (Lines 415-416)

**Point 22:**
L.388-389 – This statement is only valid for ripples where all particles can be moved. The shells were not moving, so is it surprising that the migration rate was lower? Fewer sand grain were exposed in the mixed sand-shell beds, so the answer here probably is that the ripple migrated slower because less sand was available for bedload transport.

**Response 22:** Yes, our review of the GoPro footage shows that the shells were indeed immobile. Thus, the reduced availability of the sand caused a slower migration of ripples on average. For further details, please see the time series figures under Point 15. We wanted to make the point that there are currently no good predictors of sediment transport involving shells.

**Author's changes:** (Lines 430-440)
"…~~Sediment gains and losses due to resuspension or deposition are typically absent (van den Berg, 1987), and attempts have been made to account for this by including sediment density (van Rijn, 1984, 2006). Ripple size is generally thought to scale with the thickness of the viscous sublayer (Lapôtre et al., 2017; Yalin, 1985) and does not change with velocity (Baas, 1994). At a given shields value, which is the nondimensional number that is used to calculate sediment motion, coarser-grained bedforms migrate faster than finer-grained ones (Baas et al., 2000; Lichtman et al., 2018). Yet, this clearly does not hold true forwhichIn fact~~ Nevertheless, the addition and subsequent coarsening due to shell valves and fragments dampened the ripples up to 7-fold with height, more than 2-fold in length and with an order of magnitude reduction in migration rate (bare sand vs. 50 % shells; Fig. 4a, b and d)."

**Point 23:**

L.403-416 – I do not see what this information adds to the narrative. What is meant by "the flow hat is characteristic of our study typically consists of diurnal or semidiurnal tides, instead of unidirectional flows"? Bidirectional tides are entirely different from unidirectional flows. If his study really intended to simulate tidal flows, the use of unidirectional flow experiments needs to be fully justified in the introduction and methods. Ripples in oscillatory are also entirely different, so the rather vague statements on wave-dominated environments add nothing substantial to the paper in my opinion. The other statements in this paragraph can be removed or they should be explored/ explained in much more detail, e.g. the potentially interesting sentence on L.410-412.

**Response 23:** We agree that this paragraph does not add to the core message of our experiment/study and have removed the middle section, which includes the diurnal/semidiurnal tides, how shells potentially enhancing/reducing sediment motion and how these may affect the boundary layer conditions.

**Author's changes:** (Lines 453-467):

"Care must be taken in drawing comparisons as these are dissimilar environments with entirely different causes for the armoring. As mentioned above, the shells were already immobile from the start to finish in our experimental runs, and the long-term formation/evolution processes of sand-shell beds remains inconclusive. Moreover, unlike the riverine gravel, which is closer to a spherical shape, shells are an entirely separate class of materials with biological origins. ~~Moreover, the flow that is characteristic of our study typically consists of diurnal or semidiurnal tides, instead of unidirectional flows. As of yet, it is uncertain how oscillating flow might impose further complexities on a shell-laden sandy bed. However, the relationship between shells and ripples is neither linear or even always positively correlated. Normally, in current-generated ripples, the motion is dominated primarily by flow-induced shear stresses, while immobile materials as shells enhance the turbulence in smoother beds and provide stability in rougher beds. Depending on the bed profile, shell content can either enhance or reduce sediment motion and ripple migration. Thus, our two types of experiments yield valuable information since measurements on the shifts in boundary layer conditions that occur early on are not visually detectable or quantifiable from the analyses at equilibrium.~~ Under typical unidirectional flow conditions, a higher shell % can be expected to dampen ripple development, migration and, consequently, the bedload transport. How shells might affect the hydrodynamics and bed morphology under more-complex systems and flow conditions, particularly in shallower, wave-dominated environments, remains to be investigated (e.g., under sheet or oscillatory flow conditions; Nelson et al., 2013; Precht & Huettel, 2003; Soulsby, 1997)."

**Point 24:**

L.424 – The most commonly used Wentworth scale states that 0.352 mm sand is medium-grained sand.

**Response 24:** We have revised the description of the sediment from "fine-grained" to "medium-grained" in the two instances (Lines 110 and 475).